# How do language models learn facts? Dynamics, curricula and hallucinations

**Nicolas Zucchet**[*]
ETH Zürich[†]

**Jörg Bornschein**
Google DeepMind

**Stephanie Chan**
Google DeepMind

**Andrew Lampinen**
Google DeepMind

**Razvan Pascanu**
Google DeepMind

**Soham De**[*]
Google DeepMind

## Abstract

Large language models accumulate vast knowledge during pre-training, yet the dynamics governing this acquisition remain poorly understood. This work investigates the learning dynamics of language models on a synthetic factual recall task, uncovering three key findings: First, language models learn in three phases, exhibiting a performance plateau before acquiring precise factual knowledge. Mechanistically, this plateau coincides with the formation of attention-based circuits that support recall. Second, the training data distribution significantly impacts learning dynamics, as imbalanced distributions lead to shorter plateaus. Finally, hallucinations emerge simultaneously with knowledge, and integrating new knowledge into the model through fine-tuning is challenging, as it quickly corrupts its existing parametric memories. Our results emphasize the importance of data distribution in knowledge acquisition and suggest novel data scheduling strategies to accelerate neural network training.

Large language models offer a powerful and intuitive interface to access the vast amounts of knowledge on which they were trained. The learning process acts as a lossy compression algorithm turning training data into neural network parameters, thereby implicitly determining what information is preserved within the final model. The extent and nature of this information loss depend on numerous factors, including architecture, training objective, and data distribution, dependencies that remain poorly understood. Deciphering the mechanisms underlying knowledge compression is increasingly important as these models are becoming our main gateway to human knowledge. Our work addresses this challenge by systematically investigating how language models acquire factual knowledge.

Inspired by a long history of work in neuroscience focusing on associative recall to study (Pavlov, 1927) and model (Hopfield, 1982) intelligence, we focus on a synthetic factual recall task to study in depth the learning dynamics of language models. Our contributions are:

– We adapt the synthetic task of Allen-Zhu & Li (2023), which generates artificial biographies for testing factual recall, to make it suitable for studying the formation of associative memories (Section 1).

– We find that language models learn in multiple phases on this task. They first learn overall distribution statistics, then their performance plateaus, and, finally, they acquire individual-specific knowledge (Section 2.1). Leveraging a novel attention patching technique, we demonstrate that attention-based recall circuits develop during this plateau (Section 2.2).

– We analyze the impact of data distribution properties on these dynamics, revealing that imbalanced distributions accelerate the transition through the intermediate plateau phase (Section 3) but lead to overfitting. We further demonstrate that data scheduling strategies can exploit this accelerated transition while mitigating overfitting, providing a rare example in which data curricula benefit (self-)supervised learning.

---

[*]Correspondance to nzucchet@ethz.ch and sohamde@google.com.
[†]Work done at Google DeepMind.

- We highlight the ineffectiveness of fine-tuning to incorporate new knowledge in the model (Section 4). This stems from two related factors: First, models hallucinate (overconfident predictions on unseen individuals) as soon as they acquire individual-specific knowledge. Second, associative memories stored in feed-forward layers are rapidly corrupted when training on new individuals.

# 1 An experimental setup to track knowledge over the course of learning

This study investigates the learning dynamics underlying factual recall and knowledge acquisition in language models. This presents two main methodological challenges: First, we need to isolate the model's knowledge from other abilities. Second, we want to evaluate the models over the course of learning, which requires computationally efficient measurement techniques. Following Allen-Zhu & Li (2023), we train language models on synthetic biographies that feature key properties of datasets used to train large language models. By carefully designing these synthetic biographies, we can attribute the model's ability to predict specific tokens solely to its acquired knowledge and efficiently measure its knowledge through its performance on these tokens. In this section, we first define knowledge and contrast it with memory, then describe our synthetic dataset and introduce the metrics we use to track knowledge during training, and finally, describe training details.

## 1.1 Knowledge, and how it differs from memory

For our analysis, we define knowledge to be the information a model has internally stored about its training data, abstracted from the specific form in which it was encountered. It is important to distinguish it from memorization. While knowledge involves accessing and applying information flexibly across different contexts, memorization is tied to particular training instances. In this view, knowledge can be understood as a form of memorization in a latent semantic space. For instance, knowing that Paris is France's capital represents knowledge, while remembering the exact sentence 'Paris is the capital of France' would constitute memorization. From a practical perspective, this distinction carries significant implications. Memorization can be problematic due to potential data leakage (e.g., Nasr et al., 2023), while knowledge is a more desirable property that enables models to ground their outputs in factual reality, making their responses both accurate and generalizable.

## 1.2 Synthetic biographies as a framework for studying knowledge

An important methodological challenge in our study is to isolate knowledge from other model capabilities. To address this, we adapt the synthetic biography dataset of Allen-Zhu & Li (2023), which offers several appealing properties. First, all facts in the dataset are atomic, meaning no fact can be derived from others, thus allowing us to disambiguate knowledge recall from other abilities like reasoning. Second, it is fully synthetic, which allows for precise control of the data distributional properties, while employing natural language with realistic statistics. For example, sufficient textual variation is crucial for enabling genuine knowledge acquisition rather than mere memorization (Allen-Zhu & Li, 2023), cf. Figure A in the appendix. Third, and most importantly, previous work has established that relatively small language models trained on this dataset (Allen-Zhu & Li, 2023) store and use knowledge in a similar way to large language models (Geva et al., 2021; Meng et al., 2022; Geva et al., 2023; Nanda et al., 2023b).

We summarize the data generation process in Figure 1. The dataset contains a population of $N$ randomly sampled individuals, each with a unique name and six attributes: birthdate, birthplace, university, major, company, and location. To generate a biography, we first sample an individual from the population. For each attribute, we then sample a template from a finite pool, fill it with the individual's information, and concatenate the resulting sentences in random order to form a complete biography. It should be noted that to make the dataset suitable for our study, our template generation and manipulation differ from the original setup. We detail these modifications in the next section and provide additional details in Appendix B.

**1. Create N individuals with unique names beforehand**

**James Frida Zhu**

**Birthdate:** 16/05/2042

**Birthplace:** Shanghai

**University:** Erasmus university, Rotterdam

**Major:** Statistics

**Company:** Global Dynamics

**Location:** Cairo

**2. Sample one template per attribute and fill in these templates with personal information**

James Frida Zhu's life began on March 16, 2042.

James Frida Zhu is a native of Shanghai.

James Frida Zhu received their education in Erasmus University, Rotterdam.

James Frida Zhu holds a degree in Statistics.

James Frida Zhu currently works for Global Dynamics.

James Frida Zhu's habitation is in Cairo.

**3. Create a biography by randomly permuting the generated sentences and concatenating them.**

Predicting attribute tokens is a factual recall task which measures the model's knowledge.

Figure 1: **Data generation process underlying the synthetic biography dataset we train on.** We measure the knowledge stored within these models through the loss they achieve when predicting the attribute tokens, highlighted in blue. See Section 1 for more details.

### 1.3 Measuring knowledge at scale

Measuring knowledge in language models typically relies on factual question-answering tasks (Lin et al., 2021). However, this approach is computationally intractable in our setup, as it requires repeatedly fine-tuning models to handle question-answer prompt formats throughout the training process. Our framework offers a simple solution to this challenge: by consistently placing information about the individual and attribute type before the attribute value, we transform each attribute value prediction into a factual recall task. This lets us quantify knowledge via the model's loss and accuracy on predicting attribute value tokens. We denote them as *attribute loss* and *attribute accuracy*. Formal definitions are provided in Appendix B. The attribute accuracy metric is arguably more intuitive, as it serves as a proxy for the accuracy with which the model, once fine-tuned, would correctly answer recall questions. However, we focus most of our analysis on the attribute loss due to its greater interpretive power and as it generally exhibits more consistent progression across various scales (Schaeffer et al., 2024).

Two important considerations remain to be solved: how to distinguish knowledge from memorization, and what is a meaningful baseline for the attribute loss. To address the first point, we evaluate models on unseen biographies of previously encountered individuals. We do so by randomly splitting the template pool into training and evaluation templates for each individual, with this split varying across individuals. This ensures that models are evaluated on entirely new biography formulations, thereby ruling out the possibility that strong performance stems from memorization. For the second point, we introduce a theoretical baseline corresponding to the best possible performance achievable by a model with no individual-specific knowledge. Such a model would only know the overall distribution of attribute values, and its loss would be equal to the entropy of the attribute value distribution. We refer to this value as the *no knowledge* baseline. Any model performing better than this baseline must necessarily have acquired some knowledge about specific individuals in the training distribution.

### 1.4 Language models are trained following standard recipes

While we use synthetic data in our experiments, we keep the architecture and the training protocol as close as possible to standard practices. By default, we train an 8-layer decoder-only Transformer (44M non-embedding parameters) with the same structure as in Hoffmann et al. (2022) using the AdamW optimizer (Loshchilov & Hutter, 2019) and a cosine learning rate schedule without warm-up. We tune the learning rate in all our experiments. Additional details are provided in Appendix B.

## 2 How language models acquire knowledge during learning

Our findings reveal that language models initially learn generic statistics before acquiring individual-specific knowledge. When individuals appear less frequently, such as when the population size grows, performance can plateau for extended periods between these two

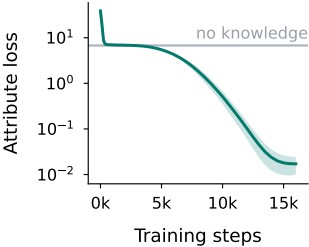 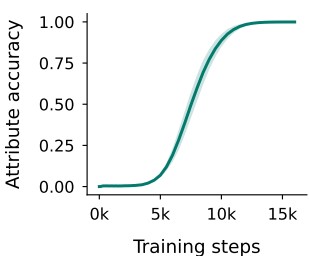 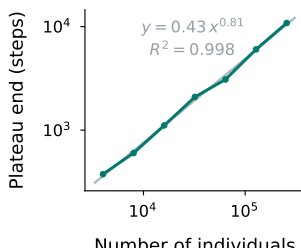

Figure 2: **Knowledge acquisition occurs in three phases.** (left) In a very short first phase, the model learns generic attribute value statistics. In the second phase, performance plateaus at a level achievable by an ideal model lacking individual-specific knowledge (this corresponds to the *no knowledge* baseline defined in Section 1.3 and a near-zero knowledge accuracy). This plateau's duration is nearly proportional to the number of individuals (right). Finally, the model learns associations between subjects and attributes: knowledge emerges as the model is trained longer (middle). Results are averaged over 5 seeds ($\pm$ std). See Section 2.1 for details.

phases. Mechanistically, we attribute this plateau to the development of attention-based circuits that enable the recall of knowledge stored within the network's parameters and modulate the learning speed of the rest of the model.

## 2.1 The three phases underlying knowledge acquisition

The temporal evolution of the attribute loss (Figure 2) exhibits a consistent three-phase pattern:

1. **Initial language understanding.** Early on during learning, the network learns the overall attribute value distribution. At the end of this phase, it performs like an optimal model without individual-specific knowledge, matching the no knowledge baseline.
2. **Performance plateaus at the edge of knowledge acquisition.** The model's performance then plateaus at the no knowledge baseline. This could be both explained from an optimization argument – the network parameters are at a saddle point of the loss landscape – or from a statistical argument – the model needs to observe multiple biographies from the same individual to discern that attribute values are specific to each individual. The plateau length grows almost linearly with the number of individuals (Figure 2, right panel), supporting the statistical hypothesis.
3. **Knowledge emergence.** The model finally enters a knowledge acquisition phase, in which it progressively learns associations between individuals and their attributes. Its knowledge accuracy progressively leaves the near-zero regime in this phase: knowledge emerges.

The pattern described above is robust across a range of hyperparameter configurations. Specifically, qualitative results remain consistent when varying learning rate, weight decay, batch size, number of individuals, model size, and even the type of sequence mixing block (attention-based vs. recurrent), cf. Figure C. This is consistent with the findings of Nichani et al. (2024), who observed a similar pattern in a theoretically tractable version of the task learned by a linear attention layer. These findings suggest that the data structure, particularly the high individual-to-attribute ratio, plays a critical role in driving this multi-phase learning behavior. Gu et al. (2025) found a similar behavior in a setup closely related to ours, and that, below some critical model size, the model does not learn any individual-specific information at all despite extensive training. We additionally discuss connections to existing work studying Transformer learning dynamics through the lens of $N$-grams in Appendix A.

## 2.2 The attention-based circuits supporting recall are created during the loss plateau

Although knowledge retrieval performance remains constant during the plateau phase, we observe the development of attention-based circuits supporting individual-specific

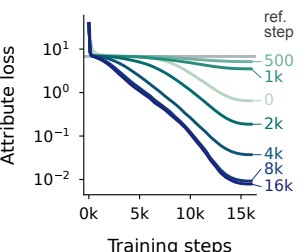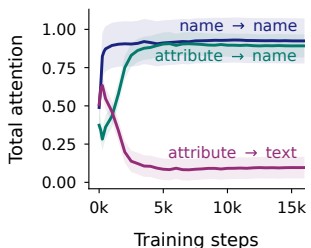

Figure 3: **The attention-based circuits supporting recall are created during the loss plateau.** (left) We design an attention patching experiment, in which we take a snapshot of a reference model at some time during its training, and use its attention patterns as a replacement for the ones of a modified model throughout its training. (middle) The more trained the reference model is, the better its attention patterns are for the modified model, and these changes mainly happen during the plateau. The very beginning of learning is an exception to this trend. This correlates with the fact that, at this time during training, the name tokens (compared to the rest of the text, which contains information about the attribute type) receive reduced attention when the first attribute value token is predicted (cf. right panel). See Section 2.2 for more details.

attribute recall. We argue that learning is considerably stalled when these circuits are under development, as errors are not properly backpropagated from attribute value tokens to name tokens, explaining the existence of the plateau. The following evidence supports this claim.

Previous work has analyzed how Transformer-based language models solve factual recall tasks similar to the one we are interested in this study. It can be summarized as follows (cf. Figure E for a visual summary). The first attention layers aggregate name tokens together to form a representation of the individual's name at the position of the last name token. This representation serves as a query for subsequent layers, particularly the multi-layer perceptrons, which effectively act as a key-value database (Geva et al., 2021; Meng et al., 2022). Individual-specific knowledge thus typically appears within the final name token's residual stream (Nanda et al., 2023b; Allen-Zhu & Li, 2023; Geva et al., 2023). The final attention layer then selects the relevant information conditioned on the attribute type and makes it available for predicting attribute value tokens when needed (Geva et al., 2023; Nanda et al., 2023b). With this last circuit established, attribute value prediction errors directly backpropagate to relevant tokens, here the name tokens. However, without it, errors are spread across irrelevant tokens, hindering learning efficiency. We therefore hypothesize that the plateau phase corresponds to the formation of these attention-based recall circuits, with learning speed directly linked to their development stage.

A consequence of that hypothesis is that a model with learned attention patterns should learn faster than one with pre-learning patterns. To test this, we design the following *attention patching* experiment (Figure 3, left). We first train a reference model and save reference checkpoints from it at various stages of training. We then restart training a model from scratch, but replace the model's attention patterns[1] with those produced from one of the reference checkpoints. This means that all the token-to-token interactions are specified by the reference model and that the modified model only learns token-wise feedforward transformations. We expect the quality of the attention patterns, measured by how easy it is to learn the task with them, provided by the reference model to progressively improve as they are taken closer to the plateau end. This is what we observe empirically (Figure 3, middle), and the plateau disappears when providing the model with learned attention patterns (i.e. post-plateau patterns). Interestingly, the attention patterns acquired early during learning (e.g., at steps 500 and 1k) are significantly worse than the pre-learning ones, likely due to initial focus on predicting the attribute value distribution conditioned on the attribute type only. This attention on attribute type tokens rather than on name tokens slows down learning, as per our argument from the last paragraph.

---

[1]We call attention patterns the softmax of the attention logits.

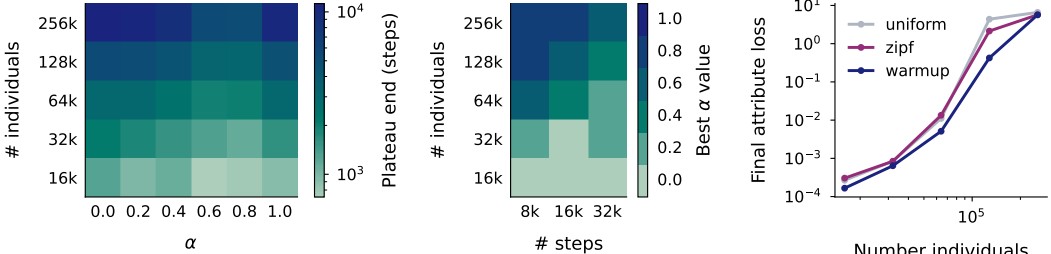

Figure 4: **Data distributional properties can speed up knowledge acquisition.** (left) The length of the plateau significantly decreases when some individuals appear more frequently (up to some extent) than other, which is here achieved by increasing $\alpha$. (middle) As a result, it is beneficial to train the model on more imbalanced distributions (higher $\alpha$ values), particularly as the number of training steps decreases or as the total number of individuals increases. This panel reports the $\alpha$ value that minimizes the final attribute loss for each number of steps and individuals. (right) Such a strategy, improves the final amount of knowledge contained in the network (purple vs. grey line). Dynamically adapting the data distribution yields even larger benefits (blue line). See Section 3 for more detail.

So far, our results show the formation of an attention-based circuit during the plateau phase, though not necessarily the specific extraction circuit we hypothesized. To investigate this further, we examine the evolution of the network's attention patterns during training for signatures of such a circuit. Specifically, we analyze the attention paid to name tokens (relative to general text tokens containing attribute type information) when predicting the first attribute value token, the primary position testing factual recall. We observe this attention increasing throughout the plateau, after being significantly lower during the initial phase where predictions align with the generic attribute value distribution (Figure 3, right panel). This offers an intuitive explanation for the qualitative differences in attention patterns observed in Figure 3 (middle). A similar analysis for the name tokens grouping circuit, characterized by high attention to name tokens from the last name token in the first layer, reveals its emergence within a few hundred steps. This may be specific to our setup, in which such a circuit is useful to better predict which template comes next. Details of this analysis are provided in Appendix D.2. Together with our attention patching results, this result suggests that the attention-based extraction circuit responsible for recall develops during the plateau phase, consistent with the findings of Ou et al. (2025). Note that our results do not rule out that other qualitative changes may occur during this phase, and they do not provide any mechanistic insights regarding how models acquire individual-specific knowledge.

## 3 Data distributional properties drive knowledge acquisition

While each individual appears with equal frequency in the preceding analysis, real-world data can be significantly less balanced. This section investigates the impact of such imbalances on the learning dynamics. We find that plateau length scales with the frequency of the most prevalent individuals (minimized in highly imbalanced distributions), whereas knowledge acquisition speed primarily depends on the frequency of the least prevalent ones (maximized in uniform distributions). This observation has two implications we confirm empirically: First, the training distribution that maximizes the final knowledge stored within the model becomes more imbalanced as the amount of knowledge within the data increases, or the network is trained for a shorter amount of time. Second, dynamically adjusting the data distribution during training allows for simultaneous minimization of plateau length and maximization of knowledge acquisition speed.

### 3.1 The trade-off underlying imbalances in the data distribution

The analysis presented in the preceding section leads to the following intuition of how the data distribution impact the plateau and knowledge acquisition phases. Assuming that the circuits that the network creates during the plateau transfer to other individuals,

reducing the number of individuals should minimize time spent in this phase, as shown in Figure 2 (right). Naturally, this assumption will break when an excessively small number of individuals is over-represented, as the model will likely overfit. As a consequence, imbalanced distributions with a small group of over represented individuals should generally shorten the plateau duration. However, this does not hold for the knowledge acquisition phase. Each individual contributes equally to the total amount of knowledge the model has, so the attribute loss on the entire population will asymptotically behave like the one on the group which is learned slowly, that is, the least frequent one. For that reason, a uniform distribution should be optimal for knowledge acquisition. As a consequence, there is a trade-off: imbalanced distributions speed up the plateau and slow down knowledge acquisition, whereas the reverse holds for more uniform distributions. The distribution that optimizes the final amount of knowledge stored within the network weights should therefore become increasingly more imbalanced as the plateau takes a larger portion of training time, i.e., when the number of training steps decreases or when the number of individuals increases.

To empirically test this intuition, we modify the individual occurrence probability to follow an inverse power law with exponent $\alpha$. This exponent controls the balance of the distribution: $\alpha = 0$ recovers the uniform distribution whereas $\alpha = 1$ corresponds to the highly imbalanced Zipf law. Figure 4 (left) reports how the number of training steps needed to escape the plateau evolves with different numbers of individuals and $\alpha$ values, when the total training budget is fixed to 16k steps. As expected, increasing $\alpha$ reduces the plateau length. However, excessive increases are detrimental, likely due to overfitting. We find that the optimal $\alpha$ minimizing plateau length lies between 0.6 and 0.8, irrespective of the population size, while the plateau length itself increases with the total number of individuals. We expect this improvement to be particularly significant when the plateau consumes a large portion of training – i.e., with large population sizes or limited training budgets. Figure 4 (middle) confirms this: the $\alpha$ minimizing final attribute loss follows the anticipated trend. Overall, the properties of the data distribution significantly impact final knowledge retrieval performance, as demonstrated in Figure 4 (right), especially when individuals appear infrequently, as is likely the case during the pre-training of large-scale models.

We are aware of three related findings. First, Allen-Zhu & Li (2023) demonstrated the benefits of "celebrities" in a similar setup to ours. Our analysis provides a partial explanation for this phenomenon. While we find that celebrities offer comparable benefits to our inverse power law distribution (cf. Appendix E), their effect may be amplified in their setup due to their use of a single biography for non-celebrities, whereas we never repeat biographies. Second, Charton & Kempe (2024) showed that repetition benefits Transformer training on arithmetic tasks. Their learning curves exhibit numerous plateaus, and sample-level repetition reduces time spent on these plateaus. Our analysis may generalize to this task, suggesting a broader principle underlying our findings. Finally, Park et al. (2024) trained Transformers on a synthetic mixture of Markov chains and found that low task diversity shortens plateau length, albeit leading to solutions that do not generalize as well out-of-distribution.

## 3.2 Data schedulers increase the final amount of acquired knowledge

The previous section revealed a trade-off in choosing a data distribution, depending on the learning phase we wish to optimize. That analysis assumed a fixed distribution throughout training. However, we can envision data distribution schedulers that dynamically adapt the distribution to the current learning phase. We confirm that this strategy can yield substantial improvements.

We implement a "warm-up" strategy: the model initially trains on a subset of individuals for a fixed number of steps, and on all individuals afterwards. Starting with a subset of individuals should shorten the plateau, while the subsequent uniform distribution maximizes knowledge acquisition once the recall circuits are well established. Indeed, we observe significant gains, particularly when the number of individuals is large and the model starts acquiring knowledge (Figure 4, right), which is a practically relevant regime. Experimental details and additional analysis are available in Appendix E.

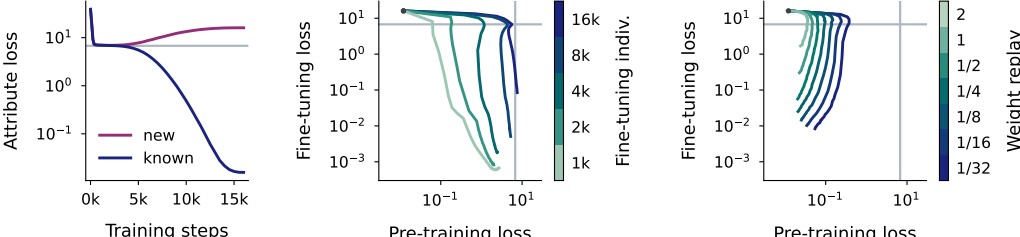

Figure 5: **Hallucinations hinder the integration of new knowledge post-training.** (left) Hallucinations (overconfidence in inaccurate predictions) appear concurrently with the knowledge acquisition, hindering subsequent adaptation to new knowledge. (middle) Fine-tuning on new individuals causes a rapid drop in performance on individuals learned during pre-training, with new knowledge acquisition being a slower process. (right) Incorporating replay of pre-training data partially mitigates the final performance drop, but not the initial decline. Grey dots in the middle and right panels indicate performance at the beginning of fine-tuning. The pre-training (resp. fine-tuning) losses are attribute losses measured on pre-training (resp. fine-tuning) individuals. See Section 4 for details.

While our experiments used a fixed warm-up duration, it could be dynamically adjusted based on observed plateau termination. This training strategy shares goals with curriculum learning (Bengio et al., 2009), but differs in that data complexity remains constant. Although preliminary and task-specific, this promising result suggests a novel strategy for accelerating training in neural networks that exhibit similar loss plateaus on learned sub-tasks. Further investigation is needed to better understand the precise conditions under which data repetition can effectively reduce time spent on performance plateaus during training.

## 4   Hallucinations hinder the integration of new knowledge post-training

This final section examines the challenge of expanding language models' parametric knowledge through post-training procedures like fine-tuning. We find that certain types of hallucinations (overconfident predictions on unseen individuals) emerge simultaneously with knowledge about individuals within the training distribution and can be detected at the population level. These hallucinations significantly impact learning dynamics on new data, requiring numerous training steps to overcome miscalibration, during which pre-existing knowledge is significantly degraded, a phenomenon reminiscent of catastrophic forgetting McCloskey & Cohen (1989). Adding replay of existing knowledge to the fine-tuning data mix only partially mitigates this issue. Overall, our findings offer an explanation for the infrequent use of fine-tuning (Ovadia et al., 2023; Jain et al., 2024) for incorporating new knowledge in the model's parameters.

### 4.1   Models start hallucinating as soon as they acquire knowledge

Before investigating fine-tuning per se, we first analyze the evolution of the model's performance on unseen individuals over the course of (pre-)training, focusing on whether it predicts attribute values following the relation-based distribution, i.e., whether it matches the no-knowledge baseline. We find that the model hallucinates, confidently predicting incorrect attribute values (Figure 5, left), but with lower overall confidence compared to predictions for individuals seen during training (Figure M). Furthermore, hallucinations emerge simultaneously with knowledge acquisition about the training individuals, suggesting they may be a counterpart to accurate predictions in current language models.

### 4.2   A large portion of the pre-training knowledge is erased during early fine-tuning

Given this uncalibrated starting point, fine-tuning on new individuals is expected to be challenging. We observe a rapid collapse in performance on pre-training data during the initial stages of fine-tuning (first few hundred steps), with very little corresponding acquisition of new knowledge (Figure 5, middle). A larger number of fine-tuning individuals intensi-

fies this effect. Extended fine-tuning eventually changes this trend, and the model starts learning about new individuals. Incorporating replay data about pre-training individuals in the fine-tuning set partially mitigates the initial collapse and facilitates the restoration of corrupted knowledge (Figure 5, right). Finally, we do not observe such degradation when fine-tuning on subsets of the pre-training data (see the corresponding analysis in Appendix F.3). These results are consistent with the findings of Gekhman et al. (2024), who found that fine-tuning on new knowledge is slow and leads to hallucinations. We complement our analysis of fine-tuning dynamics with one in which the training distribution changes regularly (Appendix F.5). These experiments confirm the findings reported here, as well as highlight that it becomes easier to learn new individuals as training progresses.

We now turn to explaining this behavior. The first hypothesis we consider is related to attention patterns. Indeed, introducing new individuals may disrupt attention patterns, requiring time to restore recall ability, while keeping parametric knowledge relatively untouched, and further training may restore them. We analyze the network's attention patterns, in both our fine-tuning (Figure O) and switching distribution (Figure X) setups, and find them to be remarkably stable, therefore mostly ruling out this hypothesis. Our second hypothesis is that introducing novel individuals creates additional key-value pairs that corrupt existing ones when being learned. To test this hypothesis, we train a one hidden layer multi-layer perceptron on a toy associative recall task (Appendix F.4). With such a model, which has no attention layers at all, we are able to reproduce the main qualitative findings of the experiments above, suggesting that changes in the feed-forward associative memories of the model we consider are the main factor explaining the model's behavior. Getting a finer understanding of these dynamics constitutes an interesting future research direction. For example, investigating how introducing independent knowledge topics affects this early catastrophic forgetting could provide some practically relevant insights.

## 5   Discussion

**On the learning dynamics of language models.**   This work reveals the existence of phase transitions, the formation of induction heads being the canonical example (Olsson et al., 2022; Garg et al., 2022; Reddy, 2024), in tasks as simple as factual recall. While the results presented above were obtained on an isolated task, we speculate that even with natural text and its multi-task nature, task-specific learning dynamics retain the same abrupt transitions and plateaus appear when the formation of a new circuit is the main bottleneck for solving a specific task. We found the time spent in the transition phase to depend more on the data distribution than the model size. Emergent abilities (Wei et al., 2022) can thus result from increased training time as models are scaled. On the data distribution side, our results have two consequences: First, they suggest the benefits of using synthetic data early in pre-training, since data used before the end of the plateau is not retained in the final model. Second, data schedulers, potentially even adaptive ones that reduce diversity when performance plateaus on a task, appear to be a promising direction to improve learning speed. Finally, our identification of changes in feedforward associative memories leading to rapid performance drops during early fine-tuning provides a simple explanation for the practically-observed ineffectiveness of fine-tuning for new knowledge (e.g., Ovadia et al. (2023); Jain et al. (2024)).

**On the learning dynamics of neural networks more broadly.**   This analysis reveals that, in a simple but relevant setting, attention-based recall circuits emerge before the formation of associative memories in feedforward layers. We hypothesize that this occurs as established circuits amplify the correlation between the inputs and backpropagated errors received by feedforward layers. Prior to circuit formation, task learning remains possible (e.g., by artificially increasing name token values), but progress is slow, performance plateaus, and generalization likely suffers. Phenomena like grokking (Power et al., 2022; Nanda et al., 2023a) may indicate that learning dynamics initially find this shortcut, before abandoning it due to regularization. The proposed credit-assignment argument for plateau formation generalizes to other architectures, as evidenced by ablations in which attention is replaced with recurrence (Figure C). Decoupling the token-mixing role of attention from other computations, as advocated by Elhage et al. (2021) for mechanistic understanding of trained

Transformers, also proves powerful for analyzing learning dynamics. This perspective offers a promising avenue for future investigations into neural network learning dynamics.

**On the role of non-uniformity and connections to developmental psychology.** A key finding of this work is the precise analysis of how imbalances in the training distribution enable faster escape from performance plateaus. While not explicitly connected to the formation of attention patterns, similar phenomena have been observed in diverse settings (Charton & Kempe, 2024; Park et al., 2024), though sometimes at the cost of limited generalization. Our data scheduling results, by adapting the data to the network's current learning phase, offer a promising path towards accelerated learning without sacrificing generalization. Intriguingly, these strategies mirror the implicit curriculum experienced by developing infants (Smith et al., 2018), arising from factors like limited mobility or frequent exposure to familiar faces, and similar strategies have proven successful in reinforcement learning (Baranes & Oudeyer, 2013; Stooke et al., 2021). As argued earlier in the discussion, repetition strengthens the signal-to-noise ratio in correlations between events (tokens in our case), facilitating faster identification of key relationships and accelerating learning. Crucially, however, (progressive) diversification remains essential for optimal generalization in later stages of learning. Our findings could therefore serve as a building block for a statistical theory of development.

## Limitations

In our study, we train relatively small language models on a synthetic factual recall task to investigate knowledge acquisition dynamics. While this approach enables precise experimental control and mechanistic insights, it presents several important limitations.

**Scale and architecture.** The 44M parameter models we train are substantially smaller than the billion-parameter models used in practice. The learning dynamics we observe may manifest differently at larger scales or with different architectural choices. However, our ablations across model sizes, architectures (including recurrent variants), and hyperparameters suggest the core phenomena are robust to these variations.

**Data.** The synthetic factual recall task, while designed to capture essential properties of knowledge acquisition, only represents one specific type of knowledge language models acquire during pretraining. Different types of knowledge could interact in sophisticated ways that are not modeled in our task. This may create extra redundancy that language models could leverage to accelerate learning. Nevertheless, our fine-tuning results already align with practically observed inefficiencies of fine-tuning for integrating new knowledge, suggesting some degree of transferability.

Despite these limitations, we believe our work provides a necessary foundation for understanding knowledge acquisition in neural language models. The mechanistic insights we derive offer concrete hypotheses that future work can validate in more realistic scenarios. We hope these findings will guide the design of more effective training strategies and data scheduling approaches for large-scale language models.

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

# Appendix

## Table of Contents

# A    Related work

## A.1    Associative memories and factual knowledge of neural networks

Associative memories have been extensively studied in neuroscience. The foundational experimental work on conditioning by Pavlov (1927) sparked extensive theoretical research into the computational mechanism underlying associative memories. This led to break-through developments like Hebb's rule (Hebb, 1949) and Hopfield networks (Amari, 1972; Hopfield, 1982), which later proved instrumental in the emergence of deep learning.

In recent years, as language models have grown increasingly powerful, research has begun examining them through the lens of associative memories. Our work directly builds on this line of research. Petroni et al. (2019) first proposed that masked language models like BERT (Devlin et al., 2018), could encode relational knowledge within their weights. This analysis was subsequently extended to autoregressive Transformer-based language models: Geva et al. (2021) demonstrated that feed-forward layers act as associative key-value memories, Meng et al. (2022) showed how this stored knowledge could be located and edited, Geva et al. (2023); Nanda et al. (2023b) revealed the functional mechanisms underlying memory recall, and Allen-Zhu & Li (2023) identified conditions under which these general mechanisms are developed. Beyond these mechanistic insights, associative memories have provided a framework for evaluating the knowledge capacity of language models. This includes empirical scaling laws (Allen-Zhu & Li, 2024) and theoretical analyses (Nichani et al., 2024) which follow from a long history of capacity analyses of Hopfield networks and related models (e.g., McEliece et al., 1987).

While the above research primarily examines parametric memories stored in network weights, recent years have also seen growing interest in the in-context associative recall capabilities of sequential modeling networks. Notable examples include the induction head in Transformers Olsson et al. (2022) and comparative studies showing that the primary difference between attention-based and attention-free language models can be attributed to their different in-context associative recall capabilities (Arora et al., 2024).

## A.2    Learning dynamics of neural networks

The study of neural network learning dynamics has deep roots in early connectionist research (e.g., Hinton, 1986; McClelland, 1995; Baldi & Hornik, 1995). It has been particularly influential in the early days of deep learning. For example, the seminal work of Saxe et al. (2014) provided fundamental insights into the role of depth in neural network training.

The recent demonstration of in-context learning capabilities in large-scale models Brown et al. (2020) has sparked renewed interest in understanding the underlying dynamics. A significant contribution to this understanding came from Olsson et al. (2022), who established a causal link between the development of induction heads and a phase transition that enables in-context learning. This mechanistic understanding has been complemented by research on the influence of training data distribution on learning trajectories and implemented algorithms. Notably, Chan et al. (2022) and Singh et al. (2024) revealed how different data distributions guide networks through distinct algorithmic solutions, either facilitating in-context learning or not. The generality of these findings was later confirmed by Park et al. (2024), who reproduced them using a simplified training distribution.

Parallel to these developments, Nguyen (2024) showed that Transformer-based language models' predictions can be effectively approximated by $N-$gram statistics, with the optimal $N$ increasing throughout training. Our work observes similar dynamics during the transition from bigram to trigram predictions during the plateau phase. However, a key distinction lies in the level of abstraction: while Nguyen (2024)'s $N-$gram associations occur directly at the token level, our work observes these sequential dependencies at a higher level of abstraction.

Finally, and most related to ours, is the study of factual knowledge acquisition in large language models from Chang et al. (2024). Their key finding is that factual knowledge acquisition occurs through accumulating small probability increases when the model encounters the same knowledge, followed by gradual forgetting when not being exposed to it.

These results are consistent with ours, although we find forgetting to be more pronounced, likely because of larger training distribution shifts. Additionally, while their findings were obtained by slightly altering the training distribution of a large language model, our work goes one step further by studying in depth the impact of training distributions on learning speed.

# B    Experimental setup

## B.1    Rationale behind our design choices

Our goal is to identify a synthetic setting, as simple as possible, where a model exhibits knowledge of its training data (that is a flexible usage of it, cf. Section 1.1, not mere memorization) using mechanisms similar to large language models. This allows precise control over the data distribution and increases the likelihood that our findings generalize to large language models.

We build upon the synthetic biography dataset and analysis of Allen-Zhu & Li (2023), as well as the mechanistic interpretability studies of Geva et al. (2023) and Nanda et al. (2023b). Geva et al. (2023) and Nanda et al. (2023b) find that pretrained large language models encode individual-specific information in their residual stream as soon as the name of the individual is encountered. Allen-Zhu & Li (2023) observe such a behavior when training models on biographies whose order is permuted every new occurrence, along with having multiple biographies for "celebrities" (see their Q-probing analysis). We hypothesize that textual variation within the biography distribution is key to achieve this behavior. Therefore, our setup ensures unique biographies by introducing 25 distinct templates per attribute type (see next section) and permuting their order. Importantly, biographies are resampled for each new sequence. Figure A demonstrates the critical importance of random ordering and the need for some textual diversity within individual biographies. While we have not ablated the total number of templates per attribute type, we believe that the overall textual diversity of the distribution is necessary for the learning of robust features. In our experiments, the individual's name is repeated in every sentence to facilitate knowledge development, though we believe later occurrences could be replaced with pronouns without significantly affecting results, as (Allen-Zhu & Li, 2023) found the benefits of full name repetition primarily in the non-permuted and not so diverse regime.

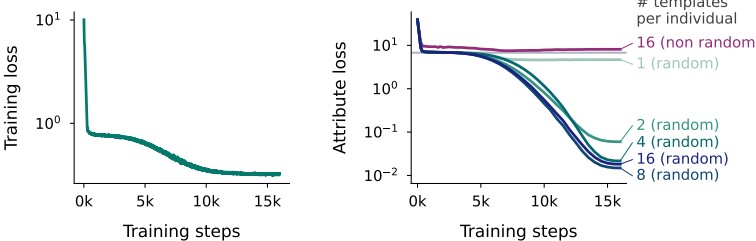

Figure A: (left) Training loss corresponding to the left and middle panels of Figure 2. (right) Ablation study demonstrating the importance of permuting the presentation order of attributes and the size of the template pool used for generating biographies. Random permutations are crucial, and some textual diversity is needed.

## B.2    Details about the biography generation process

Our data generation process largely follows Allen-Zhu & Li (2023), with a few important differences regarding how templates are generated and manipulated.

Prior to training, we generate a population of individuals, each with a full name (first, middle, and last) and six attributes: birth place, birth date, university, major, company, and current location. These are generated as follows:

- **Full name.** First and middle names are sampled from a list of 900, and last names from a list of 1,000, resulting in 810 million potential unique names. We ensure that each individual has a unique full name. These names are among the most commonly used worldwide.

- **Birth place and current location.** Sampled from a list of the 800 largest cities worldwide. Unlike Allen-Zhu & Li (2023), we do not correlate current location with the company.

- **University.** Sampled from the 200 largest universities worldwide.

- **Major.** Sampled from a list of 100 majors.
- **Company.** Sampled from the 500 companies with largest valuations.
- **Birth date.** Year, month, and day are sampled independently (between 1900-2100, January-December, and 1-31, respectively). This allows unrealistic dates (e.g., February 31st), but should not significantly impact tokenization.

These values are chosen to reasonably approximate the token distribution a large language model might encounter during pre-training.

For the sake of our analysis, the template generation requires extra care, as we want all the information needed to predict the attribute value and as we want to evaluate the model on sentences it has never seen. Such considerations were not needed in Allen-Zhu & Li (2023), so we had to adapt their setup. In our dataset, templates are generated prior to training with the assistance of a large language model, using the prompting scheme in Figure B. This yields 25 distinct templates per attribute type. As discussed in Section 1.2, we (manually) ensure that the individual's name and information identifying the attribute type precede the attribute value, allowing a model with perfect knowledge to achieve zero loss. For each individual, we pick 20 templates for training and keep 5 for evaluation, ensuring the model encounters novel template-individual combinations during evaluation, thus measuring knowledge rather than memorization. Additionally, we introduce special tokens for tags (like name or birth date) in the templates and replace these tags by the desired content only after tokenization. This way, we can make sure that, after this manipulation, the tokens coming from names or attribute values directly appear in the token sequence (e.g. "1990." will be tokenized as "[1990][.]" and not "[1990.]" as it would usually be). Such a precaution facilitates analysis as we can be sure that each of the name or attribute value tokens only contain this information, and not some unrelated semantic information about the sentence.

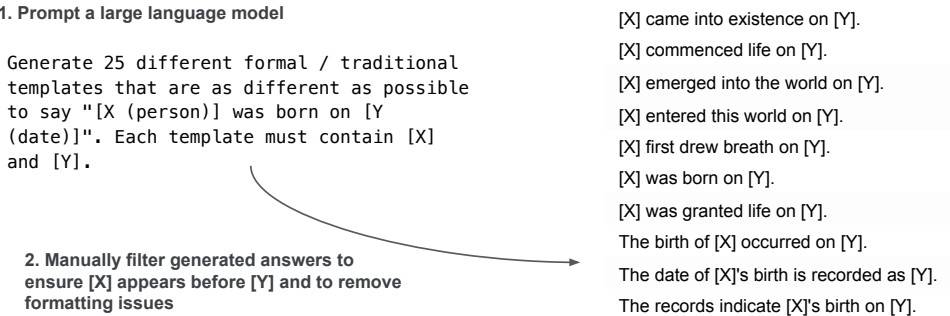

Figure B: Illustration of the template creation process. At the end of it, we have 25 different templates per attribute type.

Biography generation, for both training and evaluation, is summarized in Figure 1 and involves two steps:

1. **Individual sampling.** By default, we sample individuals uniformly at random from the population. This distribution is modified in some experiments (Section 3.1) and can be time-step dependent (Section 1.1 and Appendix F.5). Here are the detail of these different distributions:

    - **Inverse power law / Zipf distribution.** The $i$-th individual is sampled with probability proportional to $i^{-\alpha}$ with $\alpha$ an hyperparameter. Recall that we get the uniform distribution when $\alpha = 0$ and, when $\alpha = 1$, we get the Zipf distribution.
    - **"Celebrities" distribution.** A subset of the individuals of a size n_celebrities is oversampled is sampled weight_celebrities times more frequently larger than individuals outside the group. Here, n_celebrities and weight_celebrities are the two hyperparameters we vary.
    - **"Warm-up" distribution.** Training begins on a subset of indiv_warmup individuals for epochs_warmup epochs. An epoch is here defined as the number of training steps required to see as many biographies as there are individuals in the entire population (number of individuals divided by batch size steps). Once the warm-up

is over, we train on the full population. In both phases, individuals within the relevant group are uniformly sampled.

- **Sequential distribution.** The population is divided into n_groups groups and we go through these groups in increasing order n_repeats times. Individuals within each group are uniformly sampled.

2. **Biography sampling.** For a given individual, we uniformly sample templates from the appropriate pool (training or evaluation) for each attribute. We then randomize the template order and concatenate them to form a single-sequence biography.

## B.3 Architecture, optimization and metrics

In almost all our experiments, we use the 44M-parameters Transformer architecture of Hoffmann et al. (2022). It has 8 layers, uses a residual stream of dimension 512, with one hidden layer multi-layer perceptrons with 2048 hidden neurons. It has 8 heads and each head has keys and values of dimension 64. The only deviation to this architecture is the model size ablation of Figure C (lower right), in which we use a 163M-parameters (12 layers, 16 heads, dimension 896) and a 400M-parameters (12 layers, 12 heads, dimension 1536) architecture. The default hyperparameter configuration is detailed in Table 1. For most experiments, we perform a sweep over different learning rates values, selecting the best performing one based on final training loss.

The experiments in Figure 2 use 5 different seeds. However, given the low variance of the results, we preferred to use compute to explore more diverse hyperparameter configurations rather than performing our analysis on more seeds and ended up using a single seed for the rest of the experiments.

Throughout this study, we use two main metrics. The first one is the attribute loss, which is the sum of cross entropy losses measured on all attribute value tokens, which is then averaged by the number of attribute values in the sequence (always 6) and by the batch size. Importantly, we don't average by the number of tokens within each attribute value, so that the comparison to the no-knowledge baseline is easier. The second metric we use is the attribute accuracy, which is the accuracy the model gets when correctly predicting all consecutive attribute value tokens. For example, if the model always predicts correctly all attribute value tokens except the first one, it will get an accuracy of 0. This way, this metric is rather conservative, and that we are sure that for the model to be correct it has to get all the tokens requiring recall abilities (as opposed to completion abilities) right. The no-knowledge baseline is computed as the average entropy of all attribute values from all types, that is the average logarithm of the number of possible attribute values.

# C   Additional analysis of the learning dynamics (Section 2.1)

## C.1   The three phases are robust to sensible hyperparameter choices

In Figure C, we verify whether the three phases we identify are robust to sensible parameter changes. We find that they are robust to changes in learning rates, total number of individuals, weight decay values, batch size, model size and to changing the sequence mixing architecture to a recurrent one (we use the Hawk model for that (De et al., 2024)).

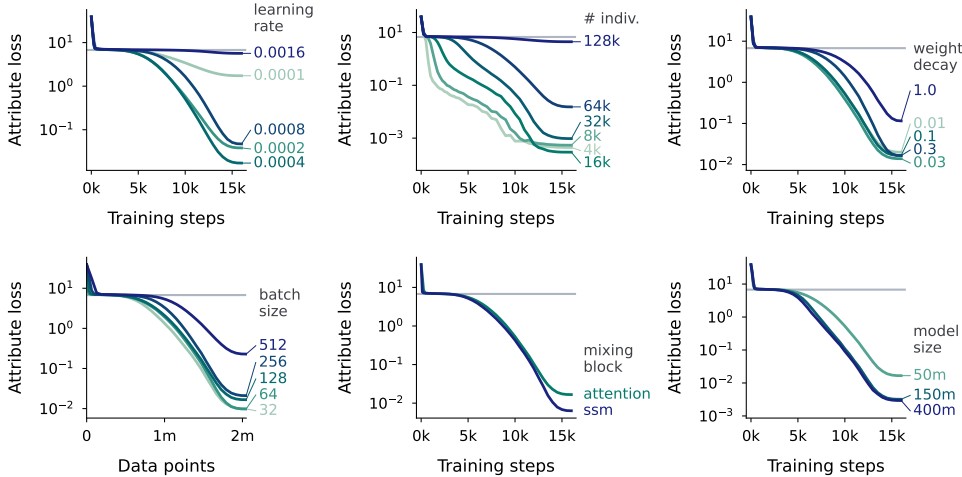

Figure C: Different hyperparameter configurations lead to qualitatively similar learning dynamics.

# D  Details of the mechanistic study and additional analyses (Section 2.2)

## D.1  Implementation of the attention patching experiment

Our attention patching experiment consists in training a reference model, then restarting training with the same initial parameters but providing the model with the attention patterns the reference model produced when seeing the same samples. Two hyperparameters control this: `reference_patching` (the number of training steps for which the reference model was trained) and `start_patching` (the step at which patching begins; before this, the modified model uses its own attention). There are two key configurations: when `start_patching` and `reference_patching` are equal, this effectively freezes the model's attention at a given point; when `start_patching` is 0, patching occurs throughout training, as presented in the main text.

We implement attention patching through a twin architecture. We initialize the modified model with the initial parameters and the reference model with the desired parameters. Both models process the input sequence layer-wise. At each layer, the reference model generates attention scores and output. The attention scores are sent to the corresponding layer in the modified model (and the output to the next layer), with gradients stopped on the attention scores to prevent training the reference model. The modified layer uses the provided attention pattern instead of its own if patching has begun. While generating all attention scores from the reference model beforehand is possible and potentially simpler code-wise, our layer-wise implementation is more memory-efficient, particularly for long sequences, and many layers and heads. That said, given that memory is not an issue in our experiments given the small size of the models, it would probably have been good enough. We always re-initialize the optimizer state to default values when starting patching. While we found some significant loss increase just after the beginning of patching with this strategy, we also found them to be smaller than without re-initializing them.

The experimental setup that we have described here is richer than the one presented in the main text as we can now vary the beginning of patching. We provide the results of this more extensive analysis in Figure D for completeness. Those results confirm our conclusions from the main text.

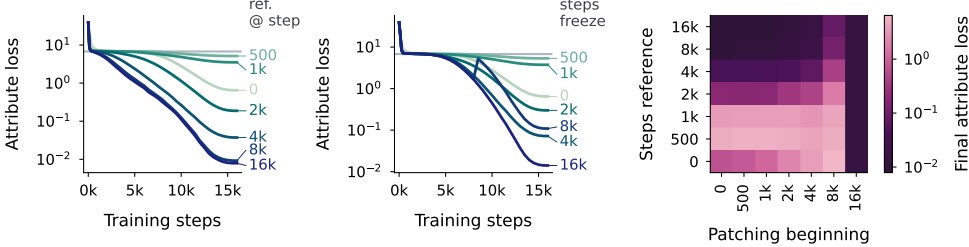

Figure D: Extended analysis of the attention patching experiment. (right) Patching starts at the beginning of learning (`start_patching = 0`). It is the same figure than Figure 3 (middle). (middle) In this experiment, the attention pattern are frozen, that is `start_patching = reference_patching`. (right) Final attribute loss when independently varying the number of steps at which we start patching and the number of steps the reference model was traning.

## D.2  Details of the attention pattern analysis

In our analysis, the model's attention patterns during learning, we examine which tokens the network attends to when processing or predicting specific tokens. This approach is motivated by prior work (discussed in Section 2.2 and visualized in Figure E) that identified specific attention-based circuits for factual recall tasks, each with distinct signatures observable through attention patterns. We focus on the following circuits and their signatures:

- **Name tokens grouping circuit.** This circuit, present in the model's first layer, groups the embeddings of name tokens to represent the individual's full name. This occurs when processing the last name token, leading to high attention on other name tokens in

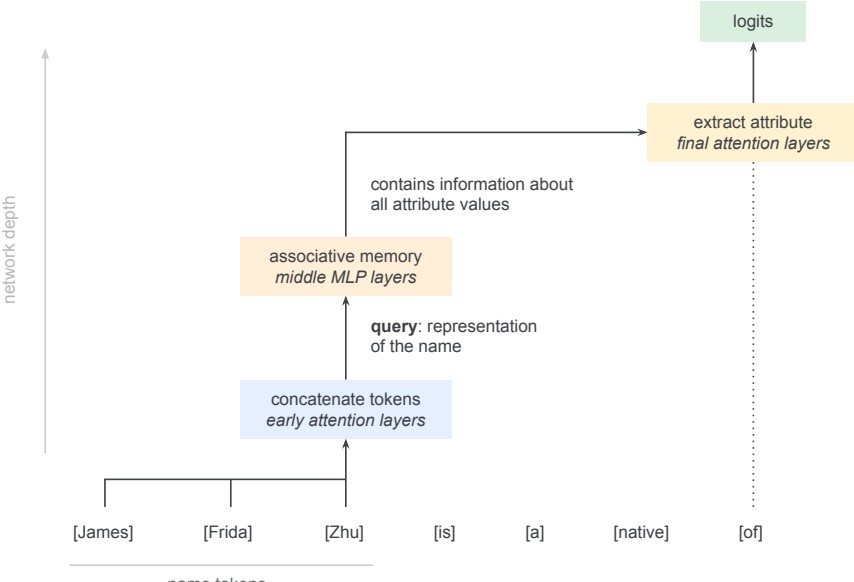

Figure E: High-level description of how Transformer-based language models solve associative recall tasks. Figure adapted from Nanda et al. (2023b). This model motivates our fine-grained attention pattern analysis (described in Section D.2) and is closely related, albeit slightly simplified, to the one of Geva et al. (2023).

> the first layer. Focusing on the last name token is crucial to differentiate this mechanism from others like name completion. The *name → name* line in the right panel of Figure 3 represents the average attention from the last name token to all other name tokens in the first layer, averaged over name occurrences (typically 6, corresponding to the number of attributes), heads, and 512 input sequences. Figure F (left) shows this quantity for all layers.

- **Extraction circuit.** The final attention layer selects and relays relevant information based on the requested attribute type. We expect high attention to name tokens (but not text tokens) when predicting the first attribute value token, once this circuit is established. Again, focusing on the first token is crucial to isolate this from completion mechanisms. The *attribute → text* and *attribute → name* lines in Figure 3 are generated accordingly. Figure F (middle left and middle right) shows the layer-wise breakdown of these quantities.

We do not include the attribute type propagation circuit highlighted in Geva et al. (2023) in this analysis to keep it simple. That said, the relatively high attention to text tokens in the first attention layers when predicting attribute values is consistent with that circuit.

As a separate observation, Figure F (right) reports the evolution of the sharpness of attention patterns in each layer, defined as the normalized entropy of the attention distribution (divided by the logarithm of the number of attendable tokens). Initially uniform, attention patterns become progressively sharper, with a slight increase in sharpness starting around the end of the plateau.

We conclude this section by discussing the limitations of this analysis. It ignores the impact of the values (e.g., attention patterns do not matter when values are equal to 0), is purely correlational, and relies on simplified mental models of the network's internal workings. Nevertheless, it enables refining the conclusions drawn from our attention patching intervention.

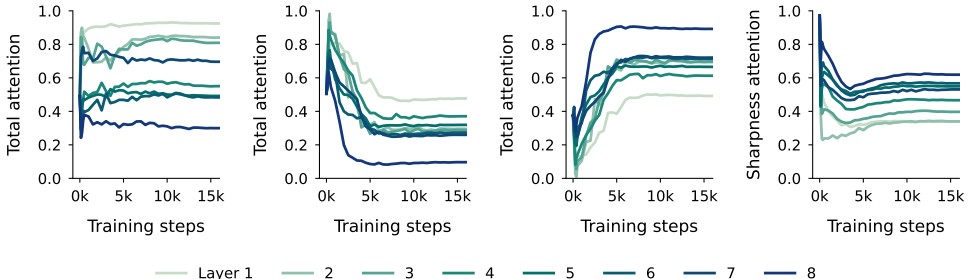

Figure F: Layer-wise analysis of the attention patterns of the model over the course of learning. (left) Attention given to the name tokens when seeing the last name tokens. (middle left) Attention given to the template tokens when predicting the first attribute value token. (middle right) Attention given to the name tokens when predicting the first attribute value token. (right) Sharpness of the attention probability distribution, defined as the entropy of the distribution divided by its mask value ($\log t$ with $t$ the index of the current token).

# E  Additional analysis for the impact of data distribution properties (Section 3)

## E.1  Learning curves for different data distributions

We here provide some examples of learning curves when modifying the training distribution. We focus on the 8k training steps, 64k individuals regime as it is one in which we observe large differences between distributions.

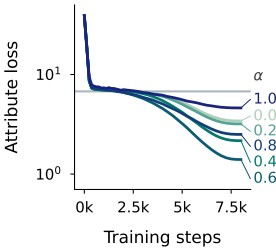

Figure G: Learning curves for the inverse power law distribution, obtained for 8k training steps and 64k individuals.

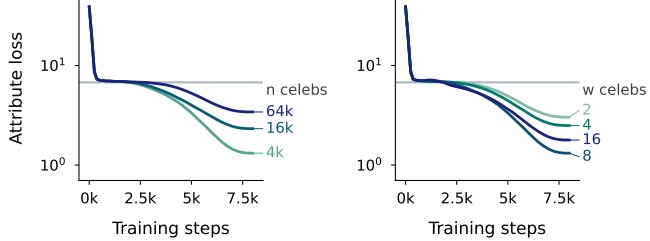

Figure H: Learning curves for the celebrities distribution, obtained for 8k training steps and 64k individuals. In the left plot, the weight for celebrities is set to 8 and in the right plot the number of celebrities is set to 4k.

## E.2  Extensive comparison of the performance of different data distributions

In Figure J, we provide a detailed version of Figure 4 (right). We vary the number of training steps and the number of individuals, and report both attribute accuracy and attribute loss. Additionally, we include the celebrities distribution in the comparison. The results stay qualitatively the same as what we reported in the main text.

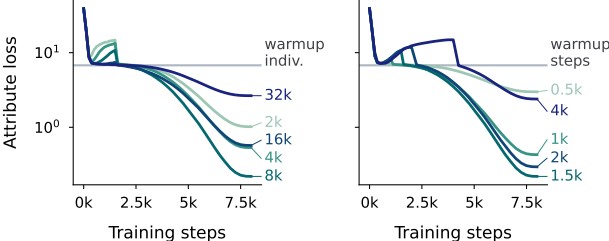

Figure I: Learning curves for the warm-up distribution, obtained for 8k training steps and 64k individuals. In the left plot, the number of warm-up steps is set to 1.5k and in the right plot the number of warm-up individuals is set to 8k.

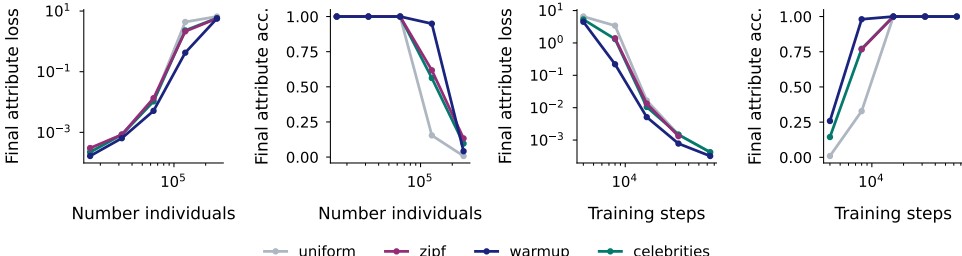

Figure J: Extensive comparison of the final performance of the model when trained on different classes of data distribution. For each class, we pick the best data distribution hyperparameter specifying the class. (left and middle left) We vary the total number of individuals, (middle right and right) we vary the number of training steps.

It should be noted that we have not tried to optimize the warm-up strategy more than through the hyperparameter grid search reported in Section G.4 as it is not the main focus of this paper, and it is possible that even better results could be achieved. For example, we only allow for changing distributions every epoch, with an epoch being defined as seeing all individuals in the training distribution. This implies that for a high number of individuals, tuning is rather coarse. Another potential improvement is to gradually increase the distribution size.

In Figure K and L, we plot the final performance of the model for different hyperparameter choices. In the low training step regime (8k steps plot in Figure K), we find the optimal hyperparameter choice to be within our grid search. However, in the large number of individuals scenario, it seems outside our range (cf. 128k individuals plot of Figure L) and in particular suggests long warmup phase with a high number of individuals. This leads us to wonder whether the benefits we observe from the warming-up could be achieved by just training on less individuals. To that extent, we run an additional run with a uniform distribution on the individual, with a population size of 83k individuals. The model reaches an attribute accuracy of 97.72%, which would become 63.26% when evaluating on 128k individuals. On the other side, the model trained with warm-ups reaches 94.92% accuracy. This refinement of our results strengthens our confidence in the benefits of warm-ups, particularly as we have not optimized the strategy beyond a relatively small hyperparameter grid search.

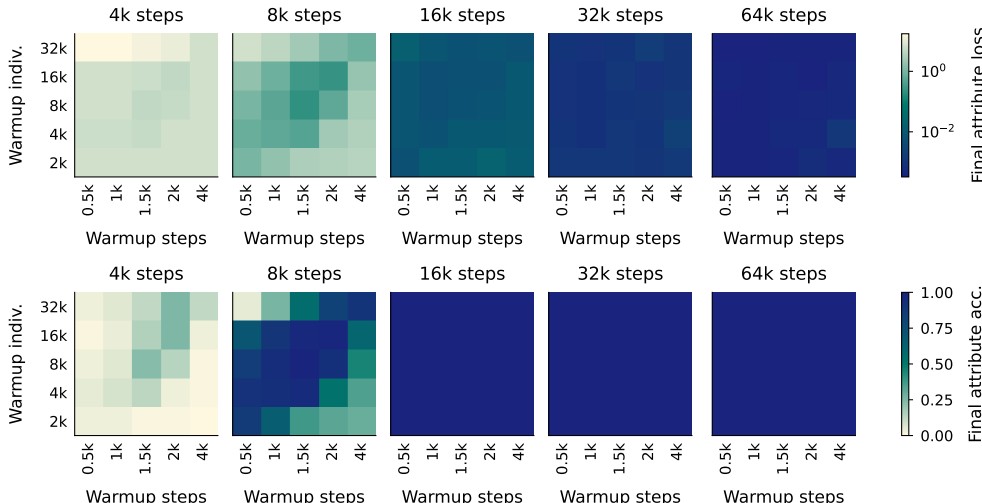

Figure K: Visualization of how final performance evolves as the hyperparameters of the warm-up distribution are changed. The total number of individuals is fixed to 64k and these plots correspond to Figure J (middle right and right).

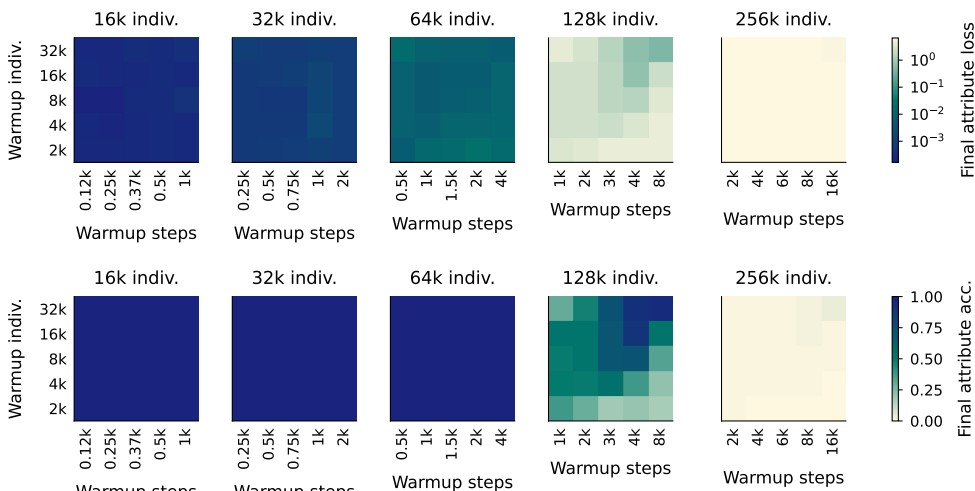

Figure L: Visualization of how final performance evolve as the hyperparameters of the warm-up distribution are changed. The total number of steps is fixed to 16k and these plots correspond to Figure J (left and middle left).

# F  Details of the fine-tuning analysis and additional experiments (Section 4)

## F.1  Experimental details

In our fine-tuning experiments, we generate `n_individuals_finetune` new individuals and use their biographies for fine-tuning. By default, we use fine-tune a model trained for 16k steps on 64k individuals, with individuals sampled uniformly (i.e., the default parameters in Table 1). In the experiments with replay, we use sampling according to the "celebrities" distribution described in Section B.2, using a weight of `weight_replay_finetune` for the 64k pre-training individuals, and a weight of 1 for the fine-tuning individuals. We use a constant learning rate of $3 \cdot 10^{-5}$ for fine-tuning, the rest of the optimizer staying at it is during pre-training.

### F.2 Hallucinations

As briefly mentioned in Section 4, our framework offers a simple way to monitor fact-conflicting hallucinations (as defined in Zhang et al. (2023)) during training. These hallucinations are characterized by overconfidence in predicting facts absent (or rare) from the training data. To assess this, we evaluate the model on held-out individuals not present in the training distribution. A hallucinating model would predict incorrect attribute values with high confidence. For these held-out individuals, the attribute loss must exceed the no knowledge baseline, with higher values indicating overconfidence (i.e., hallucinations). Figure M shows that hallucinations appear shortly after the plateau phase. However, the model remains less confident in these hallucinations than in its grounded predictions for seen individuals (see middle right and right panels). The higher probability mass on the most likely prediction and lower entropy of the predictive distribution for seen individuals suggest that hallucinations can be detected to some extent. We found these observations to be robust to change in the input distribution. For example, progressively increasing the population size did not significantly affect these metrics.

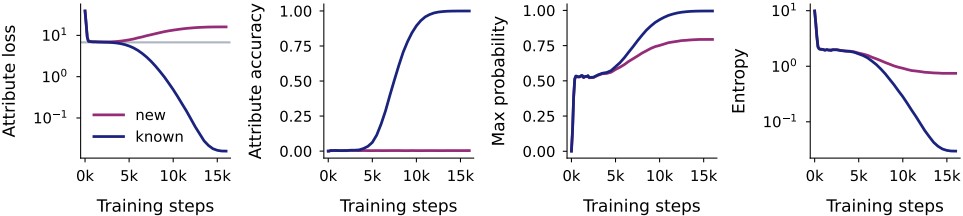

Figure M: The model starts hallucinating during training. The blue line corresponds to the model performance on seen individuals (as elsewhere in the paper) and the purple one to its performance on 16k held-out individuals. (left) attribute loss, (middle left) knowledge accuracy, (middle right) average probability of the most likely predicted token (for attribute values), and (right) average entropy of the predictive distribution (for attribute values). Overall, the model is less confident in its hallucinations than in grounded predictions.

This experiment is only a preliminary exploration of hallucinations within our setup. Future work could, for example, investigate how the confidence gap evolves when increasing the number of individuals. For instance, if we consider the network's associative memory to be a linear key-value database, increasing the number of individuals while holding the number of hidden neurons constant would bring key representations (individual names) closer, making it harder to distinguish unseen keys, thereby reducing the confidence gap and increasing hallucinations. Understanding which training individuals indirectly inform predictions for unseen individuals and what is the underlying similarity metric is another intriguing direction. Finally, on the more practical side, this framework could serve as a test-bed for methods aiming at detecting or mitigating hallucinations, as successful general-purpose methods should also be performing well in this simplified setting. We leave these investigations to future work.

### F.3 Additional analysis for fine-tuning

In the main text, we have presented results for two sets of fine-tuning experiments, one with replay and one without replay. In Figure 5, we have reported fine-tuning dynamics in a fine-tuning loss vs. pre-training loss reference frame. We here visualize the evolution of the model performance as a function of time. Importantly, these plots make it easier to remark that the drop in model performance on pre-training data occurs very early during fine-tuning (on the order of a hundred steps), and that the increase in performance on fine-tuning data is happening on a longer time-scale (see Figure N). Additionally, the accuracy plots show that the increase in loss to values close to no-knowledge baseline does not necessarily mean that all pre-training knowledge is erased as some knowledge (i.e. non-zero accuracy) about the pre-training data remains.

In Figure O, we provide the evolution of (some of) the attention scores for one of the fine-tuning experiment with no replay (4k individuals), focusing on the metrics that we have

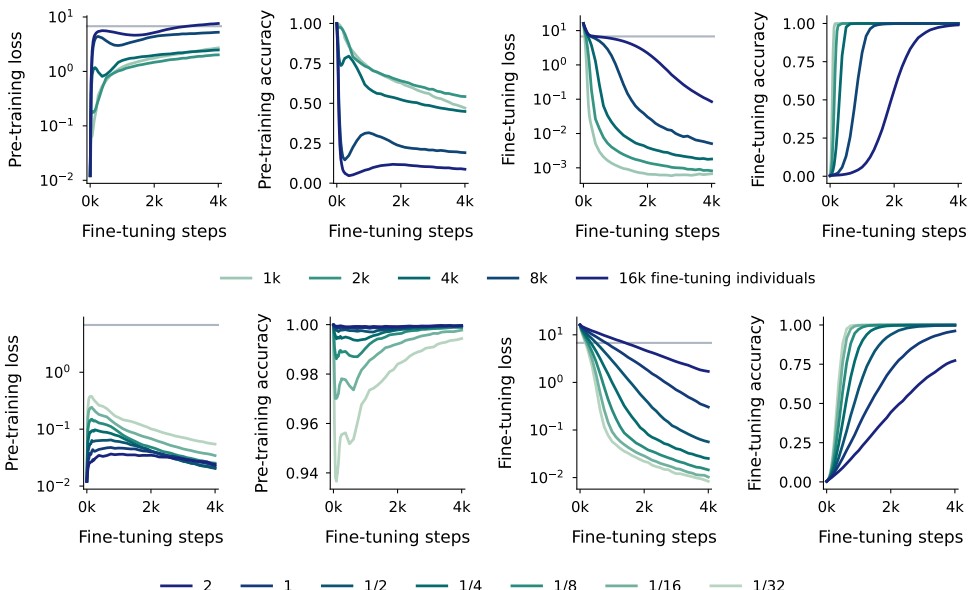

Figure N: Evolution of the performance of the model on the pre-training distribution (left and middle left panels) and on the fine-tuning distribution (middle right and right panels), as fine-tuning progresses. In the first row, there is no replay of the pre-training data during fine-tuning and we vary the number of fine-tuning individuals. In the second row, obtained for 4k fine-tuning individuals, we introduce some replay whose weight we vary (the weight corresponds to how much bigger the probability of sampling a pre-training individual is compared to one in the fine-tuning set). The data presented here is the same as the one in Figure 5 (middle and right panels), but here plotted as a function of time and including accuracy.

introduced in Section D.2 as they are most relevant to our analysis. We find attention scores to be particularly stable, which enables us to conclude that the performance drop during fine-tuning is not strongly linked to changes in attention patterns.

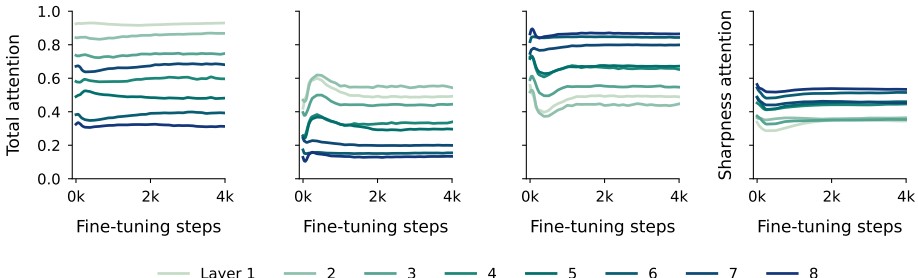

Figure O: The attention patterns remain remarkably relatively during fine-tuning. We here report the following metrics: (left) attention to name tokens when seeing the last name token, (middle left) attention to general text tokens when predicting the first attribute value token, (middle right) attention to the last name token when predicting the first attribute value token, (right) (normalized) entropy of the probability distribution defined by attention scores. The attention patterns analyzed here were obtained on the pre-training distribution; performing the same analysis on the fine-tuning distribution (not reported here) barely changes observed behaviors. See Section D.2 for the rationale behind the choice of metrics.

We complement our analysis with an experiment in which we fine-tune the model on rare, but seen during pre-training, individuals. We do so using our celebrities distribution, setting the total number of individuals to 128k, n_celebrities to 8k and weight_celebrities to 8. After training on 16k training steps, the model confidently and correctly predicts the attribute value of the celebrities but gets around 50% accuracy on non-celebrities, which

are the majority of the population. Overall, we find that fine-tuning on existing individuals does not lead to fast performance drop fine-tuning on new individuals has (Figure P). We are able to add a significant amount of knowledge in the model's weights, as the accuracy on the 120k non-celebrities goes from 50% to close to 70%. These results supporting that fine-tuning on existing individuals is net-positive are confirmed by our alternating training distribution experiment of Figure V.

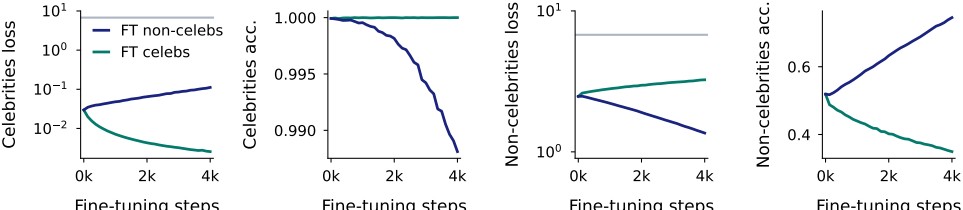

Figure P: Evolution of the model's performance when fine-tuned on rare (FT on non-celebrities line) or frequent (FT on celebrities line) individuals from the pre-training distribution. See Section F.3 for more detail.

### F.4 Reproducing fine-tuning behavior on a toy associative memory problem

In this section, we investigate whether the fine-tuning behavior observed in our language model experiments can be attributed to changes in their feed-forward associative memories. To this end, we train a single-hidden-layer multi-layer perceptron on a synthetic heterogeneous associative recall task, analogous to retrieving attribute values (e.g., university, major) given an individual's name.

Our network has 256 hidden neurons with ReLU activation. Both keys (names) and values (attributes) are represented by 64-dimensional random embeddings drawn from a normal distribution and then projected to the L2 unit sphere. The model learns to map keys to one of 30 possible values, effectively performing 30-way classification. It is learned with the cross-entropy loss and the AdamW optimizer with a cosine learning rate schedule (initial learning rate 0.03, 16$k$ steps, batch size 128, weight decay 0.01).

We pre-train the model on 8192 key-value pairs and then fine-tune it with a constant learning rate of 0.001 for 4000 steps. Fine-tuning incorporates a variable number of new key-value pairs and optionally includes replay of pre-training data. The replay weight controls the relative sampling probability of pre-training versus fine-tuning examples (e.g., a weight of 2 with 2 pre-training and 1 fine-tuning examples results in sampling probabilities of 2/5, 2/5, and 1/5).

Mirroring Section 4, we conduct two experiments:

1. Fine-tuning without replay, while varying the number of new samples (left panel of Figure Q, top row of Figure R).
2. Fine-tuning on 128 new samples and varying replay weight (right panel of Figure Q, bottom row of Figure R).

We also experimented with fine-tuning on old key-value pairs, as in Figure P, and once again observed qualitatively similar behaviors. Overall, these results provide evidence for the hypothesis that mostly feed-forward associative memories are altered during the incorporation of new knowledge through fine-tuning.

### F.5 Experiments with regular changes in training distribution

To complement our fine-tuning analysis, we consider the setup in which the training distribution is regularly changed, either to a new group of individual every change (this corresponds to varying `number_groups` and fixing `number_repeats` to 1) or alternating between two groups of individuals (this corresponds to varying `number_repeats` and fixing `number_groups` to 2). While this kind of experiment is further away from the current training paradigms of large language models and is more akin to a continual learning setup (e.g., Kirkpatrick et al. (2017)), it enables us to monitor the plasticity of the network, that is how

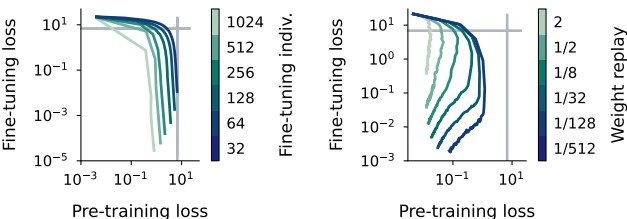

Figure Q: Equivalent of Figure 5 for the toy associative memory model. In the right plot, we set the number of fine-tuning examples to 128. See Section F.4 for experimental details.

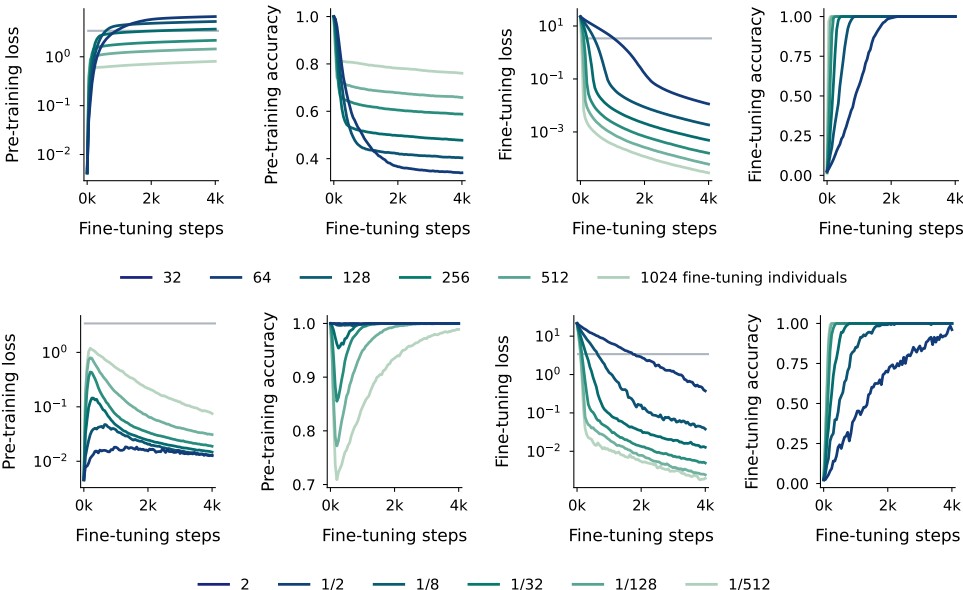

Figure R: Equivalent of Figure N for the toy associative memory model. See Section F.4 for experimental details.

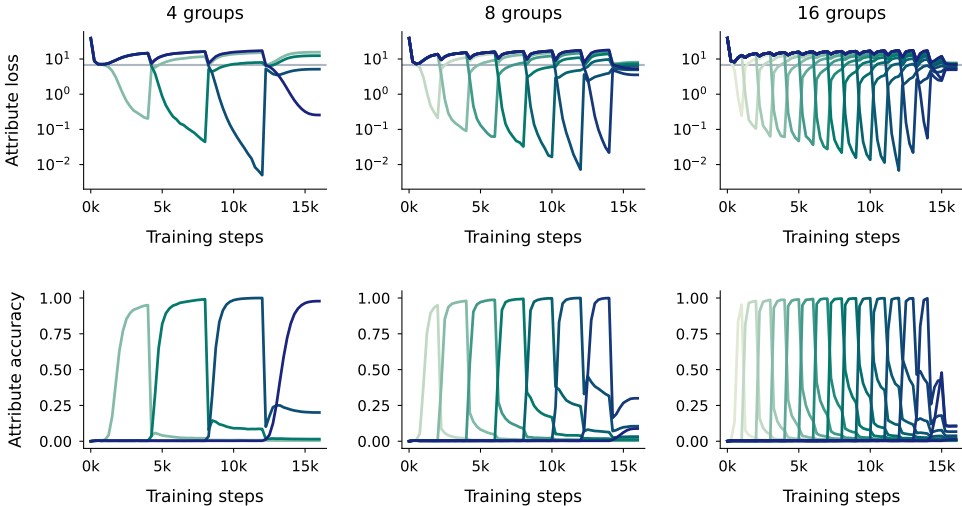

Figure S: Evolution of the model performance on different groups of individuals when it is trained sequentially on these groups. The color indicates the group of people the model is evaluated on: the darker the color, the later the network was trained on this group. For all plots the total number of individuals considered is equal to 64k. See Section F.5 for more detail.

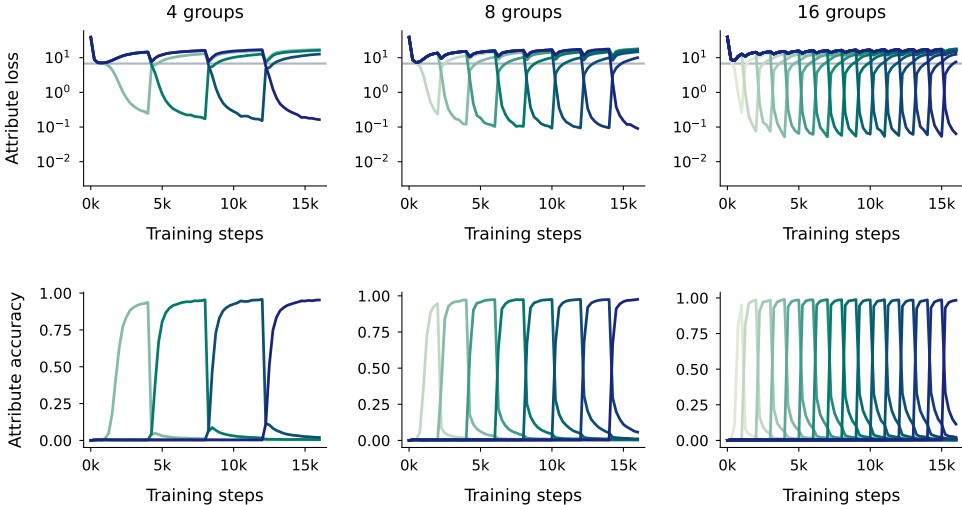

Figure T: Same as Figure S, but with a constant learning rate scheduler.

fast it learns new knowledge and forget about existing one, in a dynamic way. Overall, we find that our fine-tuning findings are robust across different stages of learning and these new results enable us to sometimes refine our conclusions.

We observe a few interesting patterns. First, it becomes increasingly easier to learn a new distribution over time (Figure S), which is likely induced by the improvement of the attention pattern quality over the course of learning (Figure W), until the cosine learning rate scheduler eventually damps down weight changes. At the same time, forgetting, measured in log loss differences, becomes more important and occurs much faster than learning, confirming the results of Figure 5. However, changing distributions multiple times during early learning dynamics when the learning rate is relatively high reduce this plasticity increase, at it partially impairs future ability of the network to learn new groups (c.f. performance drop from Figure S middle to Figure U, or Figure T). For small learning rates (i.e. at the end of training), performance on old data sometimes get improved after some initial performance drop, without it being replayed (Figure W), an intriguing phenomenon

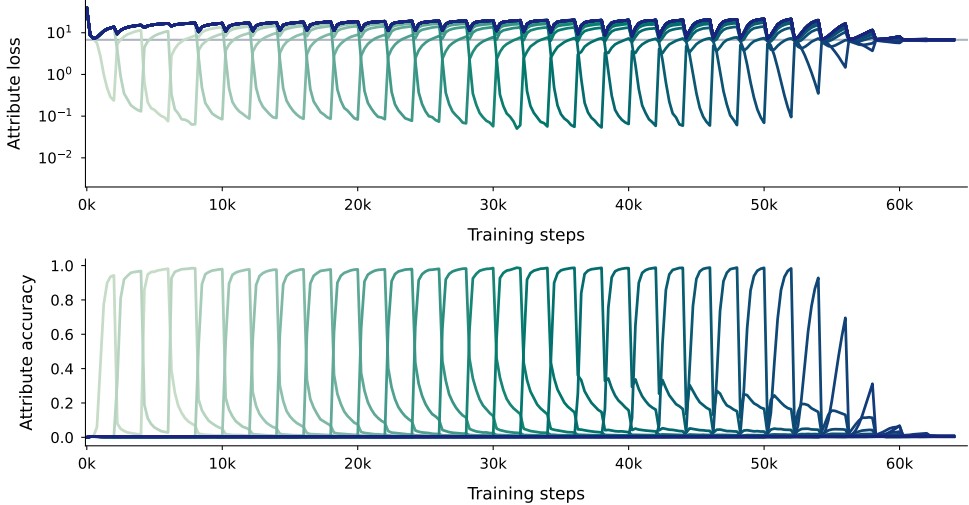

Figure U: Similar to the middle panel of Figure S (8 groups), this time when training for longer. The total number of individuals is adapted so that the size of each group remains the same as before. All other hyperparameters remain the same.

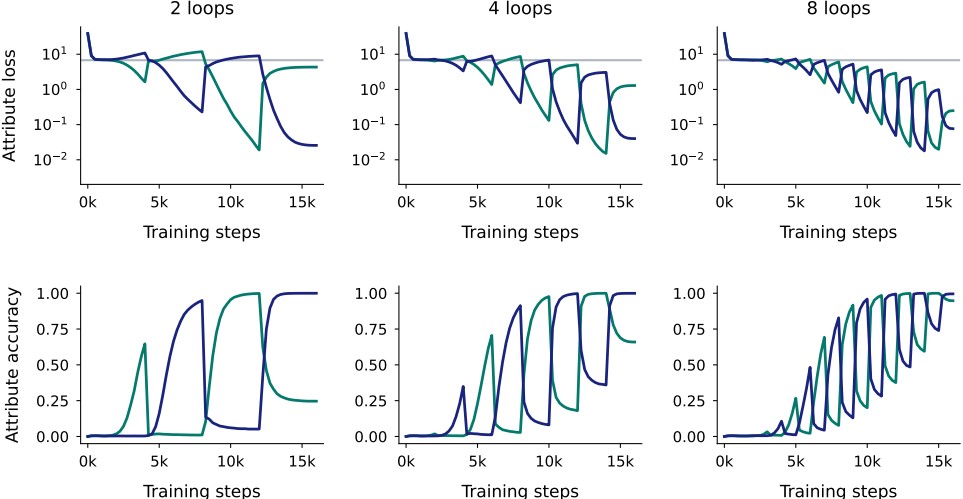

Figure V: Evolution of the model performance on different groups of individuals when it is trained alternating between two groups. The color indicates the group of people the model is evaluated on: green is for the first group, blue for the second one. For all plots the total number of individuals considered is equal to 64k. See Section F.5 for more detail.

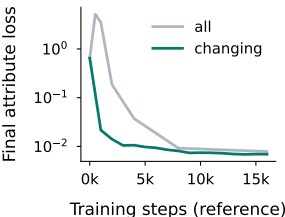

Figure W: Final attribute loss at the end of an attention patching experiment in which the model of Figure S (8 groups) serves as reference (green line), as a function of the number of steps the reference model was trained. For comparison, we include the one we obtained when training on all individuals at once (grey line), as in Figure D. We use this metric as a measure of how good the attention patterns of a given model (the reference model) at a given time are for learning and solving the task at hand. We do not observe an initial bump in performance when changing individual distribution. There are two main reasons for that: First, attention patterns get formed early on due to training occurring on less individuals. Second, the granularity that we use (2k steps) would not allow us to see such a bump here.

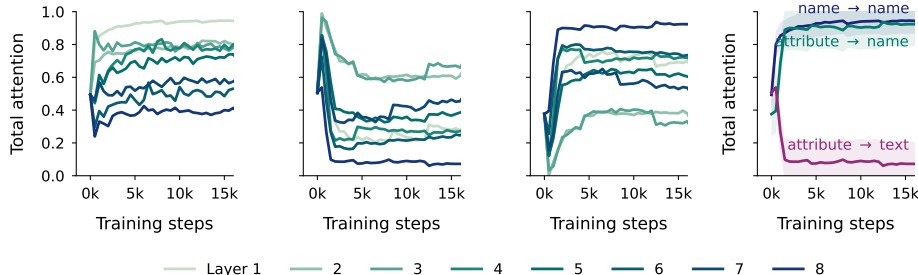

Figure X: Evolution of the attention patterns for the experiment reported in the middle panel of Figure S (one loop over 8 different groups of individuals). In the first three panels, we report (left) the average attention to name tokens when seeing the last name token, (middle left) attention to general text tokens when predicting the first attribute value token, (middle right) attention to the last name token when predicting the first attribute value token. In the right panel, we select the last layer line from the attribute to name and attribute to text plots, and the first layer line from the name to name line. See Section D.2 for the rationale behind the choice of metrics. The attention patterns are more unstable than in the constant distribution scenario (Figure F). Yet they remain relatively stable, particularly for the one highlighted in the right panel towards the end of learning, in light of the frequent distribution changes and the relatively high learning rate used for most of the training.

what we have partially observed during fine-tuning (Figure N, top row). Additionally, for very small learning rates, learning eventually becomes harder and forgetting is not catastrophic, leading to the model remembering more about the second to last group than the last one (see e.g., the 8 groups plot in Figure S). Still, too many distribution changes to unknown distributions when learning rates are small ultimately removes any knowledge from the network (e.g. Figure U).

## G   Hyperparameter configurations

### G.1   Default configuration

| Name | Value | Description |
|---|---|---|
| **Model** | | |
| parameters | 44M | Number of parameters. |
| n_layers | 8 | Number of layers. |
| n_heads | 8 | Number of attention heads per attention layer. |
| d_model | 512 | Model dimension (residual stream). |
| d_hidden | 2048 | Hidden dimension of the multi-layer perception. |
| key_size | 64 | Dimension of the key and values. |
| sequence_mixer | Attention | Which sequence mixing block we are using. |
| **Training** | | |
| training_steps | 16k | Number of training steps. |
| batch_size | 128 | Batch size. |
| lr_scheduler | cosine | Learning rate scheduler, default is cosine scheduler (no warm-up, final learning rate: $10^{-7}$). |
| lr | $4 \cdot 10^{-4}$ | Maximum learning rate. |
| weight_decay | 0.1 | Weight decay. |
| optimizer | AdamW | Optimizer (momentum parameters for the AdamW optimizer are $\beta_1 = 0.9$, $\beta_2 = 0.95$). |
| **Data** | | |
| sequence_length | 512 | Sequence length. |
| n_individuals | 64k | Number of different individuals in the population. |
| indiv_dist_train | uniform | Distribution from which we sample individuals whenever we generate a new biography (at train time). |
| indiv_dist_eval | uniform | Same as previous line, but for evaluation. |
| shuffle_templates | True | Whether to shuffle templates. |
| n_templates | 25 | Number of templates per attribute type. |
| n_templates_train | 20 | Number of templates used in training. |
| n_sequences_eval | 16k | Number of sequences for evaluation. |
| **Miscellaneous** | | |
| n_seeds | 1 | Number of seeds per hyperparameter configuration. |
| tokenizer | SentencePiece | Tokenizer, as in (Hoffmann et al., 2022). |
| vocab_size | 32k | Vocabulary size of the tokenizer. |
| checkpoints | 50 | One checkpoint every 125 steps until 2k steps, and 32 checkpoints uniformly spaced over the entire learning trajectory. |
| accelerator | Google TPUv3 | Hardware accelerator. |

Table 1: Default hyperparameters used in our experiments.

### G.2 Hyperparameters for Section 2.1

**Main experiment** (Figure 2)

| lr | $[10^{-4}, 2 \cdot 10^{-4}, 4 \cdot 10^{-4}, 8 \cdot 10^{-4}, 1.6 \cdot 10^{-3}]$ |
|---|---|
| n_seeds | 5 |

Compute time: 78 hours (train) and 154 hours (eval).

**Ablation number of individuals** (Figures 2 and C)

| lr | $[10^{-4}, 2 \cdot 10^{-4}, 4 \cdot 10^{-4}, 8 \cdot 10^{-4}, 1.6 \cdot 10^{-3}]$ |
|---|---|
| n_individuals | $[4k, 8k, 16k, 32k, 64k, 128k, 256k]$ |
| n_seeds | 5 |

Compute time: 600 hours (train) and 214 hours (eval).

**Ablation weight decay** (Figure C)

| lr | $[10^{-4}, 2 \cdot 10^{-4}, 4 \cdot 10^{-4}, 8 \cdot 10^{-4}, 1.6 \cdot 10^{-3}]$ |
|---|---|
| weight_decay | $[0.01, 0.03, 0.1, 0.3, 1]$ |

Compute time: 86 hours (train) and 29 hours (eval).

**Ablation batch size** (Figure C)

| lr | $[10^{-4}, 2 \cdot 10^{-4}, 4 \cdot 10^{-4}, 8 \cdot 10^{-4}, 1.6 \cdot 10^{-3}]$ |
|---|---|
| training_steps | $[4k, 8k, 16k, 32k, 64k]$ |
| batch_size | $[32, 64, 128, 256, 512]$ |

Compute time: 685 hours (train) and 148 hours (eval).

**Ablation model size** (Figure C)

| lr | $[10^{-4}, 2 \cdot 10^{-4}, 4 \cdot 10^{-4}, 8 \cdot 10^{-4}, 1.6 \cdot 10^{-3}]$ |
|---|---|
| parameters | $[50m, 150m, 400m]$ |

Compute time: 30 hours (train) and 89 hours (eval).

**Ablation sequence mixer** (Figure C)

| lr | $[10^{-4}, 2 \cdot 10^{-4}, 4 \cdot 10^{-4}, 8 \cdot 10^{-4}, 1.6 \cdot 10^{-3}]$ |
|---|---|
| sequence_mixer | [Attention, RG-LRU (De et al., 2024)] |

Compute time: 32 hours (train) and 15 hours (eval).

**Ablation template variety** (Figure A)

| lr | $[10^{-4}, 2 \cdot 10^{-4}, 4 \cdot 10^{-4}, 8 \cdot 10^{-4}, 1.6 \cdot 10^{-3}]$ |
|---|---|
| shuffle_templates | [True, False] |
| templates_train | $[1, 2, 4, 8, 16]$ |

Compute time: 163 hours (train) and 53 hours (eval).

### G.3 Hyperparameters for Section 2.2

**Attention patching experiment** (Figures 3 and D)

| | |
|---|---|
| start_patching | $[0k, 0.5k, 1k, 2k, 4k, 8k, 16k]$ |
| reference_patching | $[0k, 0.5k, 1k, 2k, 4k, 8k, 16k]$ |

Compute time: 73 hours (train) and 160 hours (eval).

**Attention pattern analysis** (Figures 3 and F)

| | |
|---|---|
| lr | $4 \cdot 10^{-4}$ |

Run taken from results of Figure 2.

### G.4 Hyperparameters for Section 3

**Inverse power law distribution** (Figure 4, G and J)

| | |
|---|---|
| n_individuals | $[4k, 8k, 16k, 32k, 64k, 128k, 256k]$ |
| training_steps | $[8k, 16k, 32k]$ |
| indiv_dist_train | Inverse power law |
|  - alpha | $[0, 0.2, 0.4, 0.6, 0.8, 1]$ |

Compute time: 1584 hours (train) and 521 hours (eval).

**Celebrities distribution (vary number of individuals)** (Figure H and J)

| | |
|---|---|
| n_individuals | $[4k, 8k, 16k, 32k, 64k, 128k, 256k]$ |
| indiv_dist_train | Celebrities |
|  - n_celebrities | $[4k, 16k, 64k]$ |
|  - weight_celebrities | $[2, 4, 8, 16]$ |

Compute time: 1128 hours (train) and 768 hours (eval).

**Celebrities distribution (vary number of steps)** (Figure H and J)

| | |
|---|---|
| training_steps | $[4k, 8k, 16k, 32k, 64k, 128k, 256k]$ |
| indiv_dist_train | Celebrities |
|  - n_celebrities | $[4k, 16k, 64k]$ |
|  - weight_celebrities | $[2, 4, 8, 16]$ |

Compute time: 1512 hours (train) and 541 hours (eval).

**Warm-up distribution (vary number of individuals)** (Figure 4, I, J and L)

| | |
|---|---|
| n_individuals | $[4k, 8k, 16k, 32k, 64k, 128k, 256k]$ |
| indiv_dist_train | Warm-up |
|  - indiv_warmup | $[2k, 4k, 8k, 16k, 32k]$ |
|  - epochs_warmup | $[1, 2, 4, 8]$ |

Compute time: 1944 hours (train) and 1152 hours (eval).

**Warm-up distribution (vary number of steps)** (Figure 4, I, J and K)

| | |
|---|---|
| training_steps | $[4k, 8k, 16k, 32k, 64k, 128k, 256k]$ |
| indiv_dist_train | Warm-up |
|  - indiv_warmup | $[2k, 4k, 8k, 16k, 32k]$ |
|  - epochs_warmup | $[1, 2, 4, 8]$ |

Compute time: 2808 hours (train) and 1008 hours (eval).

### G.5 Hyperparameters for Section 4

**Hallucinations** (Figure 5 and M)

| | |
|---|---|
| `lr` | $4 \cdot 10^{-4}$ |

Compute time: 3 hours (train) and 5 hours (eval).

**Fine-tuning without replay** (Figure 5, N and O)

| | |
|---|---|
| `n_individuals_finetune` | $[1k, 2k, 4k, 8k, 16k]$ |
| `lr_finetune` | $3 \cdot 10^{-5}$ |

Compute time: 17 hours (train) and 95 hours (eval).

**Fine-tuning with replay** (Figure 5 and N)

| | |
|---|---|
| `n_individuals_finetune` | 4k |
| `weight_replay_finetune` | $[2, 1, 1/2, 1/4, 1/8, 1/16, 1/32]$ |
| `lr_finetune` | $3 \cdot 10^{-5}$ |

Compute time: 26 hours (train) and 174 hours (eval).

**Fine-tuning on celebrities** (Figure P)

| | |
|---|---|
| `n_individuals` | 128k |
| `training_steps` | 16k |
| `indiv_dist_train` | Celebrities |
|   `- n_celebrities` | 8k |
|   `- weight_celebrities` | 8 |
| `lr_finetune` | $3 \cdot 10^{-5}$ |

Compute time: 6 hours (train) and 48 hours (eval).

**Sequential learning with new groups** (Figure S, T (and U))

| | |
|---|---|
| `n_individuals` | 64k (256k) |
| `training_steps` | 16k (64k) |
| `lr_scheduler` | [cosine, constant] (cosine) |
| `indiv_dist_train` | Sequential |
|   `- n_groups` | $[4, 8, 16]$ (32) |
|   `- n_repeats` | 1 |

Compute time: 26 (+13) hours (train) and 78 (+432) hours (eval).

**Sequential learning with alternating groups** (Figure V)

| | |
|---|---|
| `indiv_dist_train` | Sequential |
|   `- n_groups` | 2 |
|   `- n_repeats` | $[2, 4, 8]$ |

Compute time: 10 hours (train) and 39 hours (eval).

