# OpenReview forum: "How do language models learn facts? Dynamics, curricula and hallucinations"
_colmweb.org/COLM/2025/Conference — COLM 2025_

### Official Review · Reviewer_HLX1 · 2025-05-10

**Rating:** 8
**Confidence:** 4
**Ethics Flag:** 1

**Summary:**

How LLMs learn vast knowledge during pre-training is a critical and important problem. In this paper, the authors investigate the learning process of LLMs using a synthetic factual recall task, and 1) identifies the multi-phrase phenomenon for knowledge learning during pre-training; 2) investigates the impact of data distribution on learning dynamics; 3) reveals the ineffectiveness of fine-tuning for now knowledge learning.

**Reasons To Accept:**

1. Revealing the underlying dynamics of knowledge learning is important. And this paper designs good experiments, conducts comprehensive analysis, and reveals insightful conclusions about the fact learning process during pre-training.

2. The three findings (the multi-phrase phenomenon, the impact of data distribution, and  the ineffectiveness of fine-tuning for new knowledge) are insightful, which are valuable for future LLM studies and pre-training algorithm design.

**Reasons To Reject:**

The authors only conduct experiments on a synthetic factual recall task. Although I like this clear and simple setting, I still worry whether the mismatch between such a simple synthetic task and complicated real-world pre-training tasks may lead to biased and unreliable conclusions.

---

> ### Author Response · Authors · 2025-06-02
>
> We thank the reviewer for their strong positive assessment. We are glad they found our experiments comprehensive and our conclusions insightful. We note that the other reviewers have also mentioned the applicability of the results to more realistic scenarios as a limitation of our study; we discuss this point in detail in our overall response. We believe our work provides a necessary and solid foundation for future investigations in more complex settings.

---

### Official Review · Reviewer_EVVD · 2025-05-12

**Rating:** 6
**Confidence:** 3
**Ethics Flag:** 1

**Summary:**

This paper investigates how language models acquire factual knowledge during training using a synthetic biography dataset. It identifies a three-phase learning dynamic: (1) learning attribute distributions, (2) a plateau phase with no individual-specific knowledge, and (3) a knowledge acquisition phase. The plateau corresponds to the emergence of attention-based recall circuits. The authors also show that imbalanced training distributions can shorten this plateau and propose a curriculum to accelerate training. Finally, they examine hallucinations and show that fine-tuning on new individuals degrades existing knowledge.

**Questions To Authors:**

Line 249: The illustration seems unmatched with the figure (middle). What does the middle panel of Figure 4 intend to interpret?

**Reasons To Accept:**

- This paper proposes a systematic and reasonable way to study factual knowledge during pre-training effectively. The synthetic setup enables clean isolation of factual recall and avoids confounds from reasoning or memorization.

- The paper robustly demonstrates a three-phase training dynamic and links the plateau to recall circuit formation via attention patching. Given this insight, the paper shows how data imbalance and scheduling can speed up learning, while also providing mechanistic insight into why fine-tuning struggles to incorporate new facts.

- In summary, this paper provides an advanced understanding of how and when models internalize knowledge, which is important for both interpretability and training strategy for future work.

**Reasons To Reject:**

- Despite the systematic measurement method in this paper, the synthetic task, small models, and clean structure may not reflect real-world LLM training; the results may not scale.

- The paper would be improved if the authors could revise their presentations, re-organize their papers and provide more detailed explanations in certain sections. For example, some sections (e.g., attention patching) are dense and could be better explained; key concepts like the "no-knowledge baseline" are not emphasized upfront.

---

> ### Author Response · Authors · 2025-06-02
>
> We thank the reviewer for the positive feedback about our work. We discuss the applicability of our results to more realistic scenarios in our global response and address the specific points raised by the reviewer below:
> - **Re presentation**: We appreciate the reviewer highlighting that the attention patching section is too dense. We will incorporate additional details (currently in the appendix) to make this clearer.
> We define the *no knowledge* baseline in line 115. We will further improve the description of this baseline in the paper.
> - **Re Figure 4 middle**: The caption accurately describes the figure, although we will add additional details to the caption to make it clearer. The figure reports the alpha value that minimizes the final loss for different (training steps, number of individuals) configurations. As the number of steps decreases or the number of individuals increases, the plateau takes a larger fraction of the training and it becomes increasingly important to shorten it and thus increase alpha.
>
> We hope our responses address the reviewer concerns and invite them to consider updating their review accordingly. We remain available for additional clarifications.

---

> > ### Comment · Reviewer_EVVD · 2025-06-09
> >
> > Thank you for the detailed rebuttal. I've read your responses carefully, and I appreciate the clarifications and proposed improvements.

---

### Official Review · Reviewer_ERnd · 2025-05-12

**Rating:** 7
**Confidence:** 3
**Ethics Flag:** 1

**Summary:**

This paper conducts several experiments and presents in-depth analysis of how LLMs learn factual information. The authors work with synthetic datasets (to distinguish so-called knowledge from memorization) and work work with relatively small LLMs (so that they can actually "afford" to run all the experiments). I find the paper does provide interesting insights.

I have no problem with the synthetic dataset or the smaller LLMs. Whatever is going in terms of learning with larger LLMs trained with massive texts, this paper provides interesting observations about what is going on in the scenario the authors work on. Further, this paper seems to be a great fit for COLM; one could argue that there is not enough nlp content (actually, I would say there is no nlp at all) and that there is not substantial ml content either.

**Reasons To Accept:**

- Given that we do not know how any why LLMs learn what they learn, I find the topic of the paper timely. It is unknown whether the insights described in this paper will transfer to larger LLMs and other pretraining data (say, when pretraining data is constradictory, dynamic, uses hedging and so on), but I enjoyed reading the conclusions and analyses.

- I appreciate the detailed descriptions in English and lack of fancy (and rather useless) formulas. In particular, the description of knowledge v. memorization and baselines (106) is necessary and easy to follow.

- The authors have masterfully selected the plots to make their point in the main paper, and explain them properly. There is a ton of insights to be drawn from those plots, and it would be hard to do so without the description provided in the paper. I did not go carefully over the appendices, but they contain additional figures and they seem to have supporting materials.

**Reasons To Reject:**

I think the paper is strong, but I have to say that to a certain degree the paper is very data-driven, meaning that the authors "read aloud" what they empirically observed. There is no evidence that the claims and insights transfer to other LLMs or learning scenarios.

Likewise, despite some (unsuccessful in my opinion) effort to link their effort to neuroscience (24), there is absolutely no reason to do so; that link is not credible to me. There is no theoretical support for this link. There is no link between what they observe in LLMs and neuroscience.

---

> ### Author Response · Authors · 2025-06-02
>
> We thank the reviewer for their enthusiasm about our paper. We address their concern about generalizability in our overall response. Regarding the neuroscience connection: we do not claim a theoretical link but simply note (line 24) that understanding how brains store associative memories has historically driven important neuroscience research.

---

> > ### Comment · Reviewer_ERnd · 2025-06-03
> > **Response ack, nothing more to add.**
> >
> > Response ack, nothing more to add.

---

### Official Review · Reviewer_84Ed · 2025-05-26

**Rating:** 7
**Confidence:** 3
**Ethics Flag:** 1

**Summary:**

The paper presents experiments showing the learning dynamics of facts during LLM training. The experiments are rigorously done and have potentially impactful insights, although none of the insights have actually been tested (and are also probably out of scope for the paper to be fair). The paper is quite dense and not very reader friendly.

**Questions To Authors:**

Why did the authors not plot the regular language modelling loss as well in the training process?

In certain places in the paper, the authors refers to additional details being in the appendix but do not create a reference link for it. Makes it much easier for the reader if they can do that (eg line 124)

**Reasons To Accept:**

The paper presents some insightful findings including the phases in which facts are learnt during training, the effect of training distribution on training speed, and the effect of using fine-tuning for incorporating new knowledge.

The experimental design is rigorous.

**Reasons To Reject:**

1. All experimental results represent a toy setting, which may or may not hold for actual LLM training. I am specifically talking about how the facts about individuals are formatted. The value of an attribute of an individual never comes before the subject, which does not happen in real training. This leaves me to wonder if the plateau seen in the loss is a manifestation of this artificial constriction. In usual training, the model probably sees the attribute values appear all over.

2. The hypothesis of attention circuits being created is not completely convincing to me. The experiment steps followed by the authors to me sound similar to "distillation", where the authors provide the attention scores for a particular input from a trained model. So instead of providing internal activations, the model is provided with the attention scores (if my understanding is correct). Personally, I would always expect the plateau to go away if I get internal representations of a converged model early in the training. I am not sure if this really can be attributed to just the attention circuits forming. A concrete experiment to prove me wrong would be - Instead of providing attention scores, provide the intermediate activation vector in the MLP module just after the non-linearity. It does almost the same function as the attention scores, it takes a linear combination of the columns of the last MLP matrices. I would expect a similar set of loss plots as sen in Figure 3. Would the authors expect that or a different loss plot? What would either of those things means in terms of the authors' interpretation? Would love to hear the authors thoughts on this.

3. The paper is quite dense and could have been arranged better. The authors do list down their line of thought but at times the paragraphs feel like a dump of thoughts (this is probably a subjective opinion and can potentially be ignored)

---

> ### Author Response · Authors · 2025-06-02
>
> We thank the reviewer for their positive feedback. We address some points in our general response and respond to specific concerns below:
> - **Re name before attribute value**: One of the concerns of the reviewer is that enforcing names to be before attribute values limits the applicability of our results to realistic data. We believe that **it should not be a concern, for the reasons we explain next**. First, we argue that the ability to answer facts of the form (name, value?) is fundamentally distinct from ones of the form (value, name?) – a phenomenon known as the reversal curse (e.g., https://arxiv.org/abs/2309.12288). As a first approximation, it is therefore reasonable to consider them as **two different facts that will be acquired independently**. Incorporating reverse facts would incorporate new biography formulation and slightly change the underlying circuits, but the underlying learning dynamics should remain the same. As an additional note, always having the attribute value last makes calculating the no-knowledge baseline considerably easier.
> - **Re attention patching experiments**: First, we would like to clarify how our experiment differs from distillation to avoid any potential confusion. In distillation, the hidden representations or the attention patterns of the teacher model would serve as a soft target for the student to learn. Here, we are doing something different in the sense that we totally ignore the student attention patterns (modified model) and replace them with the teacher ones (reference model). Another way of understanding our experiment is that we fix the way information flows between tokens (specified by the reference model) and **let the modified model learn all the feedforward processing** (which includes attention value matrices, MLPs…). Second, we do overall agree with the intuition of the reviewer that providing the representations of a trained model should make learning faster. We would like to point out a couple of **nuances** that make our results non-trivial.
>   - a) With the attention patching experiment, we are assessing the **“quality”** of the attention patterns at **a certain point in time** (the reference step): the faster the network learns with these patterns, the better they are. In particular, we are interested in **when** the quality improves, which we interpret as when the attention circuits are formed. Figure 3 middle shows that the effectiveness of the patterns increases dramatically in the first 4k steps, which indicates that the formation of the attention circuits is mostly happening during the plateau (which is confirmed by Figure 3 right).
>   - b) The modifications we make to the model are rather minimal, as we only provide attention scores to the modified model. **If we directly provide activations of the reference model, learning can become extremely easy.** For example, with the experiment suggested by the reviewer, the output of the model would be a linear function of the hidden layer of the last MLP of the reference model + the value of the skip path of the last layer of the modified model. As a result, the modified model just has to learn one linear layer to match the performance of the reference model (large weights of this linear layer correspond to ignoring the influence of the skip path). Learning is thus likely to exhibit no plateau at all, but this would not give us any insight into why plateaus were occurring in the first place.
> - **Re presentation**: Please see our general response to all the reviewers on improving the readability of the paper. We will also incorporate appendix links into the main paper as suggested.
> - **Re training loss plot**: We include an example in Figure A (appendix). However, this captures abilities beyond factual knowledge (e.g., template completion); thus, we generally focus on the attribute loss which provides a more specific measure of the factual learning in question.
>
> We hope our responses address the reviewer concerns and invite them to consider updating their review accordingly. We remain available for additional clarifications.

---

> > ### Comment · Reviewer_84Ed · 2025-06-10
> >
> > If we provide the attention scores, doesn't the output of each head become a bilinear function with the input tokens and attention scores as the variables? That is much simpler than the original non-linear attention head, just like the function of the MLP would be simplified.
> >
> > If I understand the author's argument correctly here, sounds like the authors would expect similar behavior in terms of plateaus going away when the output activation after application of first MLP and nonlinearity is provided from a future model, which means that apart from attention head forming the model is also forming some other capabilities within the MLP layers? To me it sounds like the authors expect the MLP ablation to also do away with the plateau, and if that's the case, this will also give insight into why the plateau was formed. I am not sure why the authors dismiss this so easily when they say "Learning is thus likely to exhibit no plateau at all, but this would not give us any insight into why plateaus were occurring in the first place."
> >
> > The crux of my argument here is that there might be more happening during the plateau than the attention circuits being formed. Especially, something else may be going on in the MLP layer, but we don't know if that is the case or not because it was not checked in the same way. Can the authors comment on this?

---

> > > ### Author Response · Authors · 2025-06-10
> > >
> > > We would first like to clarify what we mean by attention circuits. By attention circuits, we mean all the processing that leads to the creation of the attention patterns. This includes the attention layer, but also the rest of the network upstream. In particular, the MLPs play some role in the attention circuits as they can do some preprocessing for the future attention layers to attend to the right tokens.
> > >
> > > **We agree with the reviewer that something else might be happening during this phase**; by design, our experiment does not enable us to say anything about that. We claim that attention patterns are formed during the plateau, as significant qualitative changes occur during that phase. The fact that something else might happen does not contradict this claim.
> > >
> > > Re the MLP experiment: as far as we understood the reviewer’s MLP experiment, they suggest providing representations of the final model at a given layer to the learning model. These representations would have some “information” about the parameters and attention patterns upstream. The new network therefore does not have to learn that part. Given that, we expect the plateau to remain or not, depending on the layer at which we do the patching. If we do so at the very last layer, the model just has to learn a linear map, and there is no reason to have a plateau (it is a logistic regression problem, and such problems do not typically exhibit plateaus). If we do so at the very beginning, the model has to learn everything, and the learning curves do not change. If we do so in the middle, the plateau will progressively disappear. With such an experiment, we can only assess when the model learns good representations at a given layer, which we believe provides less precise insights than the experiment we did.
> > >
> > > Re bilinearity: with the attention patterns provided, the attention layer is indeed a linear function in the input. We would like to point out that it does not always help learning, e.g. the 1k reference step line in Figure 3. That is, for the plateau to be accelerated, the network needs attention patterns that are suitable for the task, and the question our experiment answers is when this change happens.
> > >
> > > We hope that our answer clarifies our claims and we remain available for discussion.

---

> > > > ### Comment · Reviewer_84Ed · 2025-06-10
> > > >
> > > > Thanks! I believe the paper would benefit if this statement is included in it in some form if its not already there:
> > > >
> > > > "We would first like to clarify what we mean by attention circuits. By attention circuits, we mean all the processing that leads to the creation of the attention patterns. This includes the attention layer, but also the rest of the network upstream. In particular, the MLPs play some role in the attention circuits as they can do some preprocessing for the future attention layers to attend to the right tokens."
> > > >
> > > > Some of my concerns regarding the artificial construction of data (I acknowledge reading the comment about reversal curse, which to me is only partially satisfactory) and readability still persists. I would request the authors to try and make the material more reader friendly.
> > > >
> > > > However I do believe the paper will be valuable to the community.

---

### Author Response · Authors · 2025-06-02
**Global answer**

We thank all reviewers for their positive feedback on our work. We are encouraged that the reviewers found our work to address a **fundamental and timely question** about how LLMs learn factual knowledge during training (ERnd, HLX1), supported by a **rigorous and systematic experimental methodology** that cleanly isolates factual learning from confounding factors (84Ed, EVVD, HLX1). They also highlighted the **novelty of our insights** into the three-phase learning dynamics, particularly the discovery of the plateau phase corresponding to attention circuit formation (EVVD, HLX1), as well as our findings on data distribution effects and the limitations of fine-tuning (84Ed, EVVD, HLX1).

The reviewers addressed two primary concerns. All four reviewers (84Ed, ERnd, EVVD, HLX1) noted that our findings on the synthetic tasks **may not directly transfer to real-world LLM training scenarios**. Additionally, while ERnd praised the quality of our figures and explanations, other reviewers (84Ed, EVVD) pointed out that the **presentation could be improved**, as some sections are currently too dense and could benefit from better organization. We address these points below and respond to reviewers individually about the more specific points
- **Re generalizability to real-world LLMs**: All reviewers correctly noted that our use of a **synthetic dataset and smaller models represents a tradeoff**. Our approach prioritizes precision and the ability to isolate mechanisms over the complexity of real-world data. We argue that this **foundational work is crucial**; our work provides a comprehensive model of the dynamics of knowledge acquisition. We believe such models are **essential for guiding and interpreting future studies** on larger, more realistic scenarios where controlled experiments are infeasible. Our fine-tuning results already align with practically observed inefficiencies of fine-tuning for integrating new knowledge, and we hope **future work will validate our other findings in more realistic scenarios**. We will **include a Limitations section** with the points raised above.
- **Re presentation and clarity**: We appreciate the reviewers' feedback on unclear passages, particularly the attention patching section. Following these remarks, we will include more details (currently in the appendix) in the main text in the next version of the paper. We welcome further guidance from reviewers on other unclear sections and suggestions for improving presentation.

---

### Decision · Program_Chairs · 2025-07-08

**Decision:**

Accept

**Comment:**

This paper investigates how language models acquire factual knowledge during training using a synthetic biography dataset. It identifies a three-phase learning dynamic: (1) learning attribute distributions, (2) a plateau phase with no individual-specific knowledge, and (3) a knowledge acquisition phase. The plateau corresponds to the emergence of attention-based recall circuits. The authors also show that imbalanced training distributions can shorten this plateau and propose a curriculum to accelerate training. Finally, they examine hallucinations and show that fine-tuning on new individuals degrades existing knowledge.

Reasons To Accept:
* This paper proposes a systematic and reasonable way to study factual knowledge during pre-training effectively. The synthetic setup enables clean isolation of factual recall and avoids confounds from reasoning or memorization.
* The paper robustly demonstrates a three-phase training dynamic and links the plateau to recall circuit formation via attention patching. Given this insight, the paper shows how data imbalance and scheduling can speed up learning, while also providing mechanistic insight into why fine-tuning struggles to incorporate new facts.
* Overall, this paper provides an advanced understanding of how and when models internalize knowledge, which is important for both interpretability and training strategy for future work.
* Revealing the underlying dynamics of knowledge learning is important. And this paper designs good experiments, conducts comprehensive analysis, and reveals insightful conclusions about the fact learning process during pre-training.

Reasons To Reject:
* The synthetic task, small models, and clean structure may not reflect real-world LLM training; the results may not scale.
* The paper is quite dense and could have been arranged better. The authors do list down their line of thought but at times the paragraphs feel like a dump of thoughts (this is probably a subjective opinion and can potentially be ignored)